# Geometric Nonlinear Analysis of Timoshenko Beams with Variable

# <sup>2</sup> Cross-Section Using Co-rotational Formulation

- 3 Xin Guo<sup>1,2,3</sup>, Hailiang Feng<sup>1</sup>, Jiajun Hou<sup>1</sup>, Yanpei Gao<sup>4</sup>, Dongsheng Li<sup>4</sup>,Peng Guo<sup>4</sup>
- <sup>1</sup>Department of Bridge Engineering, School of Transportation Institute, Inner Mongolia University, Hohhot 010070, China
- <sup>5</sup> Inner Mongolia University Structure Testing Key Laboratory, Inner Mongolia University, Hohhot 010070, China;
- <sup>3</sup>Inner Mongolia Engineering Research Center of Testing and Strengthening for Bridges, Inner Mongolia University, Hohhot
- 7 010070, China;
- <sup>4</sup>MOE Key Laboratory of Intelligent Manufacturing Technology, Department of Civil and Environmental Engineering,
- Guangdong Engineering Center for Structure Safety and Health Monitoring, Shantou University, Shantou Key Laboratory of
- Offshore Wind Energy, Shantou 515063, China
- 11 Correspondence to: Yanpei Gao(19ypgao@stu.edu.cn.com), Dongsheng Li( lids@stu.edu.cn)
- Abstract. The geometrically nonlinear analysis of Timoshenko beam structures with variable cross-sections is a common
- challenge in engineering practice. However, traditional nonlinear analysis methods for such structures often suffer from limited
- 14 accuracy and computational inefficiency. To address these challenges, this study proposes an efficient geometrically nonlinear
- analysis framework for variable cross-section Timoshenko beams based on the co-rotational formulation. First, the novel
- 16 Timoshenko beam element with a variable cross-section, based on analytical displacement shape functions, is developed to
- enhance the computational accuracy of the co-rotational formulation. The Gaussian integration method is employed to compute
- the stiffness and mass matrices of variable cross-section elements, thereby improving computational efficiency. Then, the
- 19 tangent stiffness matrix of the variable cross-section beam element is derived based on co-rotational formulation and the
- 20 proposed variable cross-section beam element. Finally, the dedicated finite element program is developed and validated
- 21 through four benchmark examples and comparisons with experimental data from the literature. The results demonstrate that
- the proposed method achieves both high computational efficiency and accuracy in handling large deformations and nonlinear
- behavior. The proposed method is particularly suitable for analyzing structures with irregular or proportionally graded cross-
- 24 sections and demonstrates advantages over existing co-rotational approaches.
- Keywords: Timoshenko Beam, Geometric Nonlinear Analysis, Co-rotational Formulation, Variable Cross-section.

### 1. Introduction

- Beam structures are fundamental load-bearing components in various engineering disciplines, valued for their high strength,
- rigidity, and low weight. Although uniform cross-section beams have been extensively studied, modern engineering
- applications increasingly utilize non-uniform flexible beams to optimize mass distribution and enhance mechanical
- performance in structures such as wind turbine blades, robotic manipulators, and aerospace components (Xiao et.al., 2024;
- Elkaimbillah et al., 2021; Wang et al., 2014). These variable cross-section flexible beams frequently experience large
- deformations under operational loads, introducing geometric nonlinearities that invalidate classical linear beam theories based
- on small deformation assumptions. Therefore, understanding the geometric nonlinearity of flexible beam structures with non-
- uniform cross-sections is essential for accurate engineering analysis of such structures.
- Substantial research efforts have been dedicated to developing finite element methodologies for the geometric nonlinear
- analysis of flexible beams structures. The most commonly used finite element methods are the Total Lagrangian (TL)
- (Heyliger et al., 2020; Saravia et al., 2012; Marjamäki et al., 2009) and Updated Lagrangian(UL) (Greco et al., 2022; Turkalj
- et al., 2012; Kordkheili et al., 2011) formulations. While these approaches are widely adopted in commercial software due to
- their broad applicability, they have inherent limitations. Notably, these methods do not account for coordinate system changes

following beam element deformation, leading to unacceptable calculation errors when elements undergo large rotations. To 41 address this issue, an effective alternative for developing nonlinear beam elements is the co-rotational (CR) formulation. 42 Research on CR finite elements begin with the pioneering work of Wempner (Wempner, 1969), Belytschko and Hsieh 43 (Belytschko and Glaum, 1979), and Argyris and colleagues (Argyris et al., 1979). The key idea behind CR formulations is to 44 decompose the motion of a beam element into the sum of a rigid body motion and a pure deformational displacement, using a 45 local reference coordinate system that continuously rotates and translates with the element. Pioneering work by Rankin et al. 46 (Nour-Omid and Rankin, 1991; Rankin and Brogan, 1986) established a standard framework for calculating CR beam 47 formulation. Another significant contribution to CR beam theory was made by Crisfield and his collaborators (Crisfield, 1990; 48 Crisfield and Moita, 1996; Crisfield et al., 1997), who applied the CR formulation to solve various types of geometric 49 nonlinearities and proposed a consistent method for computing element equilibrium equations. Behdinan et al., 50 1998) extended the consistent CR static analysis to the dynamic analysis of beams undergoing large deflections. Hsiao et 51 al.(Hsiao et al., 1999) introduced a consistent CR total Lagrangian finite element formulation for the geometrically nonlinear 52 dynamic analysis of Euler beams with large rotations but small strain. Early CR methods used different shape functions for 53 computing elastic and inertial force vectors of the beam element, whereas Li et al. (Le et al., 2011; Le et al., 2014) adopted 54 cubic interpolations to formulate both inertia and internal local terms, and employed their new CR formulation to perform 55 nonlinear dynamic analysis of 2D and 3D beams. The computational efficacy and accuracy of CR approaches have further 56 expanded their applications across various structural systems (Moon et al., 2023; Meng et al., 2016; Wang et al., 2018; Kim et 57 al., 2022; Shen et al., 2021; Timoshenko et al., 1930). However, most existing CR formulations assume constant cross-sectional 58 properties, significantly limiting their applicability to variable cross-section flexible beam designs. 59 The increasing use of non-uniform flexible beams has driven recent research into their nonlinear behavior. The analog equation 60 method (Sapountzakis and Panagos, 2008; Sapountzakis and Panagos, 2008) has been employed for the nonlinear analysis of 61 Timoshenko beams undergoing large deflections with variable cross-sections. Yu and Zhao (Yu et al., 2024) developed a 62 viscoelastic beam element based on the absolute nodal coordinate formulation for various cross-sectional structures, where the 63 modified Kelvin-Voigt viscoelastic constitutive model was introduced to describe the large deformation of viscoelastic 64 materials. Building on this work, Yu et al. (Yu et al., 2024) further proposed an improved absolute nodal coordinate formulation 65 for analyzing the nonlinear behavior of variable cross-sections with large aspect ratios. Elkaimbillah el al. (Elkaimbillah et al., 66 2021) employed Vlasov kinematics to develop a one-dimensional finite element model for the nonlinear dynamic analysis of 67 thin-walled composite beams with open variable cross-sections. Additional studies have focused on the nonlinear behavior of 68 axially functionally graded beams with various cross-sections (Kumar et al., 2015; Ghayesh, 2018; Sınır et al., 2018; Xu et al., 69 2021). Regarding CR beam models for variable cross-sections, Nguyen and Gan (Nguyen, 2013; Nguyen and Gan, 2014) 70 employed the CR beam element to investigate the large displacement of tapered cantilever beams made of axially functionally 71 graded materials. Moon et al. (Moon et al., 2023) extended the work of Crisfield (Crisfield and Moita, 1996) on CR beam 72 elements by incorporating the fully populated and non-uniform cross-sectional stiffness matrix, expressed as a function of the 73 axial length, to develop an anisotropic CR beam model for variable cross-sections. Nevertheless, current CR methods for non-74 uniform flexible beams remain constrained by computational inefficiency and limited precision. 75 To address these limitations, this paper presents an improved CR beam model for variable cross-sections. This model 76 introduces two key innovations. First, the enhanced spatial Timoshenko beam element with variable cross-section is used to 77 obtain the tangent stiffness matrix and internal force vector, significantly improving the computational accuracy of the CR 78 method. Second, it eliminates the need to calculate the moment of inertia by evaluating the parameters of each cross-section, 79 thereby enhancing the computational efficiency of variable cross-section beams. The proposed CR model enables geometric 80 nonlinear analysis of variable cross-section beams with irregular and proportionally varying cross-sections. The remainder of 81 this paper is organized as follows: Section 2 develops the improved stiffness and mass matrices for the variable cross-section 82 beam element, based on analytical displacement shape functions. In Section 3, geometric nonlinear analysis of variable cross-

87

88 89

98

103

section beams is formulated using the CR formulation. Section 4 presents a comparative analysis of computational results,

including experimental data and numerical simulations, to validate the applicability and accuracy of the proposed method. The

main conclusions of this investigation are thereafter summarized in Section 5.

### 2. The improved spatial Timoshenko beam element with variable cross-section

The CR method enables the use of linear Timoshenko beam elements to derive the tangent stiffness matrix in the global coordinate system. Typically, interpolated shape functions are employed to construct the beam element. However, most of these shape functions approximate beam displacements, which introduces truncation errors and decreases computational accuracy. In this section, an improved Timoshenko beam element with a variable cross-section is proposed to improve computational accuracy by employing analytical displacement shape functions for bending deformation. The specific process is outlined below.

As illustrated in Fig. 1, a beam with variable cross-section is considered. The beam element has a total length L, with the coordinate origin at the left end. The x-axis is aligned with the longitudinal direction, while the y- and z- axes align with the principal axis of the cross-section. Typically, the displacement at any point within the spatial beam element is represented by  $\{u, v, w, \theta_x, \theta_y, \theta_z\}$ , where u is the axial displacement along the x axis, v and w are the transverse displacements along the y and z axis, respectively, and z axis, respectively, and z axis, respectively. The cross-section parameters are defined: where z is the width, z is the thickness, z is the cross-sectional area, z and z are the moments of inertia about the z-axis, respectively.

Figure 1. Variable geometric properties in a tapered beam

Define  $k_y$  and  $k_z$  as the cross-sectional non-uniformity coefficients along the y and z axes, respectively, E as the elastic modulus, G as the shear modulus, and J as the moment of inertia. Substituting the constitutive relations and geometric equations of the Timoshenko beam into the equilibrium equations yields:

$$\begin{cases} \frac{\partial}{\partial x} \left[ E I_y \frac{\partial \theta_y}{\partial x} \right] = k_z G A \left( \frac{\partial w}{\partial x} + \theta_y \right) \\ k_z G A \left( \frac{\partial^2 w}{\partial x^2} + \frac{\partial \theta_y}{\partial x} \right) = 0 \end{cases}$$
 (1)

The relationship between transverse displacement and bending displacement is given by:

$$109 w = w_b - \frac{El_y}{k_z GA} \frac{\partial^2 w_b}{\partial x^2} + \frac{El_y}{k_z GA} \frac{\partial^2 w_b}{\partial x^2} | x = 0, (2)$$

where subscripts b denoting contributions from bending deformation respectively.

Similarly, the analytical solution of transverse displacement v satisfying the boundary conditions can be obtained as:

$$v = v_b - \frac{EI_Z}{k_VGA} \frac{\partial^2 v_b}{\partial x^2} + \frac{EI_Z}{k_VGA} \frac{\partial^2 v_b}{\partial x^2} | x = 0,$$
 (3)

### © Author(s) 2025. CC BY 4.0 License.

- Similar to the traditional Timoshenko beam element, the displacements in the u and  $\theta_x$  are interpolated linearly. While the
- transverse displacements  $v_b$  and  $w_b$  are interpolated using cubic polynomial, and their expressions are given by:

$$\begin{cases} u(x) = c_1 x + c_2 \\ \theta_X(x) = c_{11} x + c_{12} \\ v_b(x) = c_3 x^3 + c_4 x^2 + c_5 x + c_6 \end{cases}$$

$$(4)$$

$$w_b(x) = c_7 x^3 + c_8 x^2 + c_9 x + c_{10}$$

In general, the strain vector of a spatial Timoshenko beam element is expressed as:

$$\begin{cases} \boldsymbol{\varepsilon} = \left[\varepsilon_{x}, \gamma_{y}, \gamma_{z}, \gamma_{x}, \varepsilon_{y}, \varepsilon_{z}\right]^{\mathrm{T}} \\ = \left[\frac{\partial u}{\partial x}, \frac{\partial v}{\partial x} - \theta_{z}, \frac{\partial w}{\partial x} + \theta_{y}, \frac{\partial \theta_{x}}{\partial x}, \frac{\partial \theta_{y}}{\partial x}, \frac{\partial \theta_{z}}{\partial x}\right]^{\mathrm{T}} \\ = \boldsymbol{\varepsilon}_{\alpha} + \boldsymbol{\varepsilon}_{\beta} \end{cases}$$

- where  $\boldsymbol{\varepsilon}_{\alpha} = \left[\frac{\partial u}{\partial x}, \frac{\partial v}{\partial x}, \frac{\partial w}{\partial x}, \frac{\partial \theta_x}{\partial x}, \frac{\partial \theta_y}{\partial x}, \frac{\partial \theta_z}{\partial x}\right]^{\mathrm{T}}$  and  $\boldsymbol{\varepsilon}_{\beta} = \left[0, -\theta_z, \theta_y, 0, 0, 0\right]^{\mathrm{T}}$ . By combining Eqs. (4) and (5), the expressions for the
- displacement and rotation vector  $\mathbf{u}(x)$  of the beam can be obtained as follows:

$$122 u(x) = A(x)c, (6)$$

- where the matrix A(x) represents the displacement-rotation coefficient matrix with respect to the shape function coefficient
- vector  $\mathbf{c} = \{c_1, \dots, c_{12}\}^T$ . Taking the derivative of Eq. (6) yields:

$$\begin{cases} d\mathbf{u}(x) = \left\{ \frac{\partial \mathbf{u}}{\partial x}, \frac{\partial \mathbf{v}}{\partial x}, \frac{\partial \mathbf{w}}{\partial x}, \frac{\partial \theta_{x}}{\partial x}, \frac{\partial \theta_{y}}{\partial x}, \frac{\partial \theta_{z}}{\partial x} \right\}^{\mathrm{T}}, \\ d\mathbf{u}(x) = d\mathbf{A}(x)\mathbf{c} \end{cases}$$
 (7)

- Based on the boundary conditions at x=0 and x=L, the relationship between the shape function coefficients and the nodal
- displacements can be derived and expressed in matrix form as follows:

$$128 H(x)c = d, (8)$$

where H(x) is the coefficient matrix of the shape function coefficients. The nodal displacement d is expressed as:

$$\mathbf{d} = \{u_1, v_1, w_1, \theta_{x1}, \theta_{y1}, \theta_{z1}, u_2, v_2, w_2, \theta_{y2}, \theta_{y2}, \theta_{z2}\}^{\mathrm{T}}, \tag{9}$$

By substituting Eq. (9) into Eqs. (6) and (7), the following expressions are obtained:

$$u(x) = A(x)H(x)^{-1}d$$
, (10)

$$d\mathbf{u}(x) = d\mathbf{A}(x)\mathbf{H}(x)^{-1}\mathbf{d}$$
, (11)

The relationship between the strain and nodal displacements of the element is then given by:

$$\begin{cases} \boldsymbol{\varepsilon}_{\alpha} = d\boldsymbol{u}(x) = d\boldsymbol{A}(x)\boldsymbol{H}(x)^{-1}\boldsymbol{d} \\ \boldsymbol{\varepsilon}_{\beta} = \boldsymbol{T}_{N}\boldsymbol{u}(x) = \boldsymbol{T}_{N}\boldsymbol{A}(x)\boldsymbol{H}(x)^{-1}\boldsymbol{d} \\ \boldsymbol{\varepsilon} = \boldsymbol{\varepsilon}_{\alpha} + \boldsymbol{\varepsilon}_{\beta} = [d\boldsymbol{N}(x) + \boldsymbol{T}_{N}\boldsymbol{N}(x)]\boldsymbol{d} = \boldsymbol{B}(x)\boldsymbol{d} \end{cases}$$
(12)

- where  $N(x) = A(x)H(x)^{-1}$ ,  $dN(x) = dA(x)H(x)^{-1}$ , B(x) is the strain-displacement matrix, and  $T_N$  satisfies the following
- relationship:

- By numerically integrating over the length L of the beam, the element stiffness matrix  $\mathbf{K}_e$  and mass matrix  $\mathbf{M}_e$  of the variable
- cross-section Timoshenko beam element are formulated as:

$$\begin{cases}
\mathbf{K}_e = \int_0^L \mathbf{B}(x)^{\mathrm{T}} \mathbf{K}_{cs}(x) \mathbf{B}(x) dx \\
\mathbf{M}_e = \int_0^L N(x)^{\mathrm{T}} \mathbf{M}_{cs}(x) N(x) dx
\end{cases} \tag{14}$$

Define J as the moment of inertia. The sectional stiffness matrix  $K_{cs}(x)$  for a variable cross-section beam is expressed as:

$$\mathbf{K}_{CS}(x) = diag[ES(x), k_y GS(x), k_z GS(x), GJ(x), EI_y(x), EI_z(x)],$$
 (15)

- Directly evaluating the integrals in Eq. (14) for variable cross-sections is often computationally intensive. Therefore, in this
- study, Gaussian quadrature is introduced to efficiently compute the element stiffness and mass matrices of the variable cross-
- section beam:

$$\begin{cases}
\mathbf{K}_e = \sum_{i=1}^n \frac{L}{2} \omega_i \mathbf{B}(x_i)^{\mathrm{T}} \mathbf{K}_{cs}(x_i) \mathbf{B}(x_i) \\
\mathbf{M}_e = \sum_{i=1}^n \frac{L}{2} \omega_i \mathbf{N}(x_i)^{\mathrm{T}} \mathbf{M}_{cs}(x_i) \mathbf{N}(x_i)'
\end{cases}$$
(16)

- where n is the number of Gaussian integration points,  $\omega_i$  and  $x_i$  are the corresponding weight coefficients and integration
- nodes, respectively.
- The stiffness and mass matrices of the cross-section are determined based on the relevant parameters of the cross-section.
- Considering the diverse forms of cross-sections, a general formula is provided here to handle the cross-sectional parameters
- of variable cross-section beams with a certain taper.
- Assuming that the aspect ratio of the variable cross-section beam remains constant, i.e.

$$\frac{b_r}{b_l} = \frac{h_r}{h_l},\tag{17}$$

- where,  $b_r$  and  $h_r$  are the width and thickness of the cross-section at the right end, and  $b_l$  and  $h_l$  are the width and thickness
- at the left end. Under this assumption, the cross-sectional parameters at any arbitrary point along the beam can be expressed
- as:

$$\begin{cases} h(x) = k_1 x + f_1 \\ b(x) = k_2 h(x) = k_2 (k_1 x + f_1) \\ S(x) = p_1 b(x) h(x) = p_1 k_2 (k_1 x + f_1)^2 = k_3 (k_1 x + f_1)^2 \\ l_1(x) = p_2 b(x) h(x)^3 = p_2 k_2 (k_1 x + f_1)^4 = k_4 (k_1 x + f_1)^4 \end{cases}$$
(18)

- The calculation of the cross-sectional parameters for each cross-section requires solving for the corresponding coefficients  $f_l$
- and  $k_i$  (i = 1,3,4). The transition coefficients  $k_2$ ,  $p_1$  and  $p_2$  do not need to be solved. This can be achieved by solving using
- the relevant parameters of the cross-section at both ends of the beam. For the fixed end of the beam, when x=0, we have h=0
- $h_l$ ,  $S = S_{max}$ ,  $I_y = I_{ymax}$ . when x=L, we have  $S = S_{min}$ ,  $I_y = I_{ymin}$ . By substituting the known parameters of the beam at
- both ends into Eq. (18), we obtain:

$$164 k_1 = h_1 \left( \sqrt{\frac{S_{min}}{S_{max}}} \right) / L$$

$$k_3 = S_{max}/h_1^2 \tag{19}$$

- $k_4 = I_{v max}/h_1^4$
- When the aspect ratio of the structure is variable, the width and thickness of the cross-section are mutually independent, By
- measuring the maximum thicknesses  $h_{lmax}$  and  $h_{rmax}$  of the cross-sections perpendicular to the y-axis at both ends of the
- structure, the expression for the cross-sectional parameters at any point within the unit can also be derived.
- Once the relevant coefficients are obtained, they can be substituted into the coordinates of the Gaussian integration points to
- calculate the cross-sectional parameters. By substituting the cross-sectional parameters into Eqs. (14) and (15), the element
- stiffness matrix  $K_e$  and the element mass matrix  $M_e$  of the variable cross-section Timoshenko beam element can be obtained.

### 3.Co-rotational formulation

The co-rotational formulation stands out by extracting the elastic deformation displacements from the overall displacements, thus predefining the projection relationship. The motion of the beam element from its initial state to the final deformed state is decomposed into rigid body motion and pure deformation. The rigid body motion component encompasses the rigid translation and rotation in the local reference coordinate system. Therefore, the core challenge of the co-rotational formulation lies in handling the coordinate transformation between different frames, thereby establishing the relationship between pure deformation and the overall deformation.

### 3.1Definition and transformation of the reference coordinate system for spatial beam elements

For the spatial two-node beam element, the reference coordinate system is defined as shown in Fig. 2. The unit orthogonal vectors  $E_i$ , i = 1,2,3, represent the global reference system of the beam element, which remains fixed and unchanged. The unit orthogonal vectors  $E_i^h$ , i = 1,2,3, represent the local reference system of the beam element after rigid body motion, which continuously translates and rotates with the beam element. The local reference system  $E_i^q$ , i = 1,2,3 represents the original coordinate system of the beam element before deformation. Additionally, the vectors  $e_i^1$  and  $e_i^2$ , define the cross-sectional reference system of the two nodes (1 and 2) of the beam.

Figure 2. Beam kinematics and coordinate systems

First, the rigid rotation of the local coordinate system  $\mathbf{E}_{i}^{h}$  is addressed. The rigid rotation matrix  $\mathbf{R}_{r}$  represents the transformation matrix from the reference system  $\mathbf{E}_{i}$  to  $\mathbf{E}_{i}^{h}$ , and its expression is given by:

$$\mathbf{192} \quad \mathbf{R}_r = \begin{bmatrix} r_1 & r_2 & r_3 \end{bmatrix} \tag{20}$$

The vector  $r_1$  is computed as the line connecting node 1 and node 2 of the beam element before and after deformation:

$$r_1 = \frac{S_2^g - S_1^g}{I}$$
, (21)

where  $s_i^g$  represents the coordinates of node i in the global reference system after rigid rotation. The length l of the beam after deformation can be obtained by  $l = ||s_2^g - s_1^g||$ .

- The directions of the remaining two axes are determined by introducing an auxiliary vector  $\mathbf{q}$ . The auxiliary vector serves two
- main purposes: (1) to solve the rigid rotation matrix in the global coordinate system; (2) to determine the differential
- relationship between the rigid rotation angle and the total displacement of the structure. Initially, the direction of q aligns with
- the local coordinate axis  $E_2^q$ . After deformation of the beam element, the determination of the auxiliary vector  $\mathbf{q}$  is related to
- the transformation of the local reference system:

$$202 q_i = \mathbf{R}_i^g \mathbf{R}_0 [0 1 0]^T, i = 1, 2, (22)$$

$$q = \frac{1}{2}(q_1 + q_2),$$
 (23)

- where  $\mathbf{R}_1^g$  and  $\mathbf{R}_2^g$  are the orthogonal matrices corresponding to the directions of the end nodes  $\mathbf{e}_i^1$  and  $\mathbf{e}_i^2$ , respectively.  $\mathbf{q}_1$
- and  $q_2$  are the directions of the left and right end reference systems of the local reference system  $E_2^q$  after rigid rotation.  $R_0$
- denotes the initial orientation of the local coordinates, and q represents the direction of the local reference system  $E_2^q$  after
- rigid rotation.
- By combining Eqs. (21), (22), and (23), the expressions for the remaining two components of the orthogonal matrix  $\mathbf{R}_r$  can
- be obtained

$$r_3 = \frac{r_1 \times q}{\|r_1 \times q\|} r_2 = r_3 \times r_1,$$
 (24)

- The local rotation matrix of the coordinate axis is defined as  $\bar{\mathbf{R}}_i$ , and the transformation from  $\mathbf{E}_i$  to  $\mathbf{e}_i^1$  and  $\mathbf{e}_i^2$  can be
- expressed as follows:

$$R_r \bar{R}_i = R_i^g R_0, i = 1,2,$$
 (25)

Since  $\mathbf{R}_r^T \mathbf{R}_r = I$ , Eq. (25) can be transformed as follows:

$$\tilde{\mathbf{R}}_{i} = \mathbf{R}_{T}^{T} \mathbf{R}_{i}^{g} \mathbf{R}_{0,i} i = 1,2, \tag{26}$$

Thus, the local rotation angles can be obtained as follows:

$$\tilde{\boldsymbol{\vartheta}}_i = \log(\tilde{\boldsymbol{R}}_i), \tag{27}$$

- 3.2Transformation of displacement vectors between the local and global coordinate systems
- The global displacement vector of the beam element is defined as  $P_g^g$ , and the displacement vector in the local coordinate
- system after removing rigid body deformations is denoted as P<sub>1</sub>. By utilizing the rotation framework described in the previous
- section, the local displacement  $P_l$  is obtained by subtracting the rigid body displacement from the total displacement  $P_g$ . The
- local internal force vector  $f_l$  and the tangent stiffness matrix  $K_l$  in the local coordinate system are computed through the
- transformation relationship between the two. The expression of the internal force vector  $\mathbf{F}_g$  in the global coordinate system
- can be derived by balancing the internal virtual work in the global and local systems:

$$V = \delta \mathbf{P}_l^{\mathrm{T}} f_l = \delta \mathbf{P}_q^{g \mathrm{T}} \mathbf{F}_q, \tag{28}$$

The variations of the displacement vectors  $P_g^g$  and  $P_l$  can be expressed as follows:

$$\delta \mathbf{P}_{l} = [\delta \bar{\mathbf{u}} \quad \delta \bar{\mathbf{b}}_{1}^{\mathrm{T}} \quad \delta \bar{\mathbf{b}}_{2}^{\mathrm{T}}]^{\mathrm{T}}, \tag{29}$$

$$\delta \mathbf{P}_{g}^{g} = \begin{bmatrix} \delta \mathbf{u}_{1}^{g^{\mathrm{T}}} & \delta \mathbf{\theta}_{1}^{g^{\mathrm{T}}} & \delta \mathbf{u}_{2}^{g^{\mathrm{T}}} & \delta \mathbf{\theta}_{2}^{g^{\mathrm{T}}} \end{bmatrix}^{\mathrm{T}}, \tag{30}$$

- where,  $\delta \bar{\theta}_{i}$ , (i = 1,2) represents the variation of spatial rotation angles in the local coordinate system after considering rigid
- body deformations, and  $\partial \theta_i^g$  (i = 1,2) represents the variation of spatial rotation angles in the global coordinate system.
- The variation of the transformation matrix involves the formation of a new matrix composed of rotational angles:

$$\delta \tilde{\mathbf{R}}_{i} = \delta \tilde{\boldsymbol{\theta}}_{i} \tilde{R}_{i}, \tag{31}$$

- where the superscript tilde denotes the skew-symmetric matrix corresponding to a vector. A new local coordinate system,
- denoted as  $P_a$ , is defined based on Eqs. (29) and (31).

$$\mathbf{P}_{a} = [\bar{\mathbf{u}} \quad \bar{\mathbf{\theta}}_{1}^{\mathrm{T}} \quad \bar{\mathbf{\theta}}_{2}^{\mathrm{T}}]^{\mathrm{T}},\tag{32}$$

- Let  $f_a$  represents the internal force vector corresponding to  $\delta P_a$ , and  $K_l$  denotes the transformed local stiffness matrix  $K_e$
- obtained in Section 2 of this paper, which is converted to a 7-degree-of-freedom system. The transformation matrix between
- vectors  $P_a$  and  $P_l$  can be obtained through the transformation relationship of their respective stiffness matrices. The final
- conversion of  $K_l$  to  $K_a$  can be expressed as follows:

$$\mathbf{K}_{a} = \mathbf{B}_{l}^{\mathrm{T}} \mathbf{K}_{l} \mathbf{B}_{l} + \mathbf{K}_{h}, \mathbf{K}_{h} = \begin{bmatrix} 0 & 0_{1 \times 3} & 0_{1 \times 3} \\ 0_{3 \times 1} & \mathbf{K}_{h1} & 0_{3 \times 3} \\ 0_{3 \times 1} & 0_{3 \times 3} & \mathbf{K}_{h2} \end{bmatrix},$$
 (33)

- The matrix  $B_l$  can be directly obtained by rotating the vector. The expressions for  $K_{h1}$  and  $K_{h2}$  are derived from the
- following equation:

$$\frac{\partial}{\partial \hat{\boldsymbol{\rho}}} [\boldsymbol{T}_s^{-T} \boldsymbol{v}] = \frac{\partial}{\partial \hat{\boldsymbol{\sigma}}} [\boldsymbol{T}_s^{-T} \boldsymbol{v}] \frac{\partial \hat{\boldsymbol{\sigma}}}{\partial \hat{\boldsymbol{\rho}}} = \frac{\partial}{\partial \hat{\boldsymbol{\sigma}}} [\boldsymbol{T}_s^{-T} \boldsymbol{v}] \boldsymbol{T}_s^{-1}, \tag{34}$$

$$T_s(\Phi) = \frac{\sin \varphi}{\varphi} I + \left(1 - \frac{\sin \varphi}{\varphi}\right) e e^{\mathrm{T}} + \frac{1}{2} \left(\frac{\sin(\varphi/2)}{\varphi/2}\right)^2 \widetilde{\Phi}, \tag{35}$$

- where v represents the bending moment acting on the two ends of the internal force vector in the local coordinate system, e
- is the unit vector corresponding to the angle vector,  $K_{h1}$  and  $K_{h2}$  correspond to  $\bar{\vartheta}_1$  and  $\bar{\vartheta}_2$  in Eq. (34). Consequently, the
- differential relationship between the rotational vector in the local coordinate system and the displacement vector in the global
- coordinate system can be derived as follows:

$$250 \qquad \begin{bmatrix} \delta \bar{\boldsymbol{\theta}}_1 \\ \delta \bar{\boldsymbol{\theta}}_2 \end{bmatrix} = \begin{pmatrix} \begin{bmatrix} 0 & \boldsymbol{I} & 0 & 0 \\ 0 & 0 & \boldsymbol{I} \end{bmatrix} - \begin{bmatrix} \boldsymbol{G}_{\boldsymbol{\theta}}^T \\ \boldsymbol{G}_{\boldsymbol{\theta}}^T \end{bmatrix} \end{pmatrix} \boldsymbol{E}^T \delta \boldsymbol{P}_g^g = \boldsymbol{P} \boldsymbol{E}^T \delta \boldsymbol{P}_g^g, \tag{36}$$

- where  $G_{\theta} = \frac{\partial \theta_r^e}{\partial P_{\theta}^g}, E = diag[R_r \ R_r \ R_r \ R_r].$
- Thus, the relationship between  $\delta P_a$  and  $\partial P_g^g$  can be obtained as follows:

$$\delta \mathbf{P}_a = \mathbf{B}_a \delta \mathbf{P}_g^g, B_a = \begin{bmatrix} \mathbf{r} \\ \mathbf{P} \mathbf{E}^T \end{bmatrix},$$
 (37)

where  $\mathbf{r} = [-\mathbf{r}_1^T \quad \mathbf{0}_{1\times 3} \quad \mathbf{r}_1^T \quad \mathbf{0}_{1\times 3}]$ . The matrix  $\mathbf{G}_{\theta}$  in Eq. (36) is related to  $\delta \boldsymbol{\theta}_r^e$ .

$$\delta \widetilde{\boldsymbol{\theta}}_{r}^{e} = \boldsymbol{R}_{r}^{T} \delta \boldsymbol{R}_{r}, \delta \boldsymbol{\theta}_{r}^{e} = \begin{bmatrix} -\boldsymbol{r}_{3}^{T} \delta \boldsymbol{r}_{3} \\ -\boldsymbol{r}_{3}^{T} \delta \boldsymbol{r}_{1} \\ \boldsymbol{r}_{2}^{T} \delta \boldsymbol{r}_{1} \end{bmatrix}, \tag{38}$$

The expression for  $r_1, r_2, r_3$ , and  $\delta r_1$  can be easily obtained. As for  $\delta r_3$ , it is related to  $\delta q$  according to Eq. (23):

$$\delta \mathbf{q} = \frac{1}{2} (\delta \mathbf{R}_{\gamma} + \delta \mathbf{R}_{\gamma}) \mathbf{R}_{0} [0 \quad 1 \quad 0]^{\mathrm{T}} = \frac{1}{2} (\delta \widetilde{\boldsymbol{\theta}}_{1}^{g} \mathbf{q}_{1} + \delta \widetilde{\boldsymbol{\theta}}_{2}^{g} \mathbf{q}_{2}), \tag{39}$$

- The expression of the matrix  $G_{\theta}$  can be obtained through Eq. (39) and  $G_{\theta} = \frac{\partial \theta_{\theta}^{e}}{\partial P_{q}^{g}}$ . The detailed derivation can be found in
- reference (Crisfield, 1990). Eq. (37) yields the relationship between the force vector in the global coordinates and the internal
- force vector in the local coordinates.

$$\mathbf{261} \qquad \mathbf{F}^g = \mathbf{B}_a^{\mathrm{T}} \mathbf{f}_a,\tag{40}$$

Similarly, by considering the variation of the force vector in the global coordinates in Eq. (37), it can be obtained as follows:

$$\begin{cases} \delta \mathbf{F}^{g} = \mathbf{B}_{a}^{T} \delta \mathbf{f}_{a} + \delta \mathbf{r}^{T} \mathbf{f}_{a1} + \delta (\mathbf{E} \mathbf{P}^{T}) \mathbf{m} \\ \mathbf{m} = [f_{a2} \quad f_{a3} \quad f_{a4} \quad f_{a5} \quad f_{a6} \quad f_{a7}]^{T'} \end{cases}$$
(41)

- where  $f_{ai}(i=1,\cdots,7)$  represent the components of the force vector  $\mathbf{f}_a$ . In conclusion, the tangent stiffness matrix in the
- global coordinate system can be obtained as follows:

$$\begin{cases}
\mathbf{K}^{g} = \mathbf{B}_{a}^{\mathrm{T}} \mathbf{K}_{a} \mathbf{B}_{a} + \mathbf{K}_{m} \\
\mathbf{K}_{m} = \mathbf{D} f_{a1} - \mathbf{E} \mathbf{Q} \mathbf{G}_{\theta}^{\mathrm{T}} \mathbf{E}^{\mathrm{T}} + \mathbf{E} \mathbf{G}_{\theta} \mathbf{a} \mathbf{r}'
\end{cases} \tag{42}$$

where:

$$D = \begin{bmatrix} d & 0 & -d & 0 \\ 0 & 0 & 0 & 0 \\ -d & 0 & d & 0 \\ 0 & 0 & 0 & 0 \end{bmatrix}, d = \frac{1}{l} (I - r_1 r_1^{\mathsf{T}}),$$
 (43)

$$\mathbf{Q} = \begin{bmatrix} \tilde{\mathbf{Q}}_1 \\ \tilde{\mathbf{Q}}_2 \\ \tilde{\mathbf{Q}}_3 \\ \tilde{\mathbf{Q}}_4 \end{bmatrix}, \mathbf{a} = \begin{bmatrix} 0 \\ \eta(f_{a2} + f_{a5})/l - (f_{a3} + f_{a6})/l \\ (f_{a4} + f_{a7})/l \end{bmatrix},$$
 (44)

$$\mathbf{P}^{\mathsf{T}}\boldsymbol{m} = [\boldsymbol{Q}_{1}^{\mathsf{T}} \quad \boldsymbol{Q}_{2}^{\mathsf{T}} \quad \boldsymbol{Q}_{3}^{\mathsf{T}} \quad \boldsymbol{Q}_{4}^{\mathsf{T}}]^{\mathsf{T}},\tag{45}$$

By utilizing the obtained tangent stiffness matrix, the difference in the global force vector can be calculated. The iterative process is employed to gradually converge the results towards the exact solution. The computational flowchart of nonlinear deformation in variable cross-section beam is illustrated in Fig. 3.

Figure 3. Flowchart of nonlinear deformation in the variable cross-section beam

### 4.Applications

282

This section presents comparative analysis between the proposed co-rotational Timoshenko beam model with variable cross-section and existing benchmark results to validate its accuracy. The validation is carried out in three stages. First, the simple beam models with the constant cross-section are simulated to verify the proposed beam model with geometric nonlinearity. Second, the proposed co-rotational model is applied to a beam with variable cross-section and evaluated against both analytical solutions and numerical results from the literature, thereby confirming the capability of the proposed model in handling non-uniform geometries. Finally, a frequency analysis is conducted on a variable cross-section beam, and the computed results are compared with experimental measurements and published data to further demonstrate the capability of the developed beam element for dynamic analyses.

### 4.1Application on constant cross-section beam element

## 4.1.1 Large deformation analysis of spatially pre-bent cantilever beams subjected to concentrated loads

A 45° cantilever circular arc beam with a radius of R=100m is subjected to a vertical concentrated load F of magnitude 300N

at its free end as shown in Fig. 4 below.

Figure 4. Pre-bent cantilever beam

292

298 The beam is divided into 8 elements, and the detailed cross-section properties of the beam are provided in Reference (Nguyen and Gan, 2014). Table 1 presents a comparative analysis of the displacements at the free end of the beam in the x, y, and z directions, as computed by the proposed method, the HAWC2 software, and the analytical solution.

Table 1 shows that the obtained large deformations from the developed co-rotational beam model in the x and y directions are -12.08m and -7.10m, respectively. Compared with the results obtained using HAWC2, the proposed approach improves the computational accuracy by 0.3% in the x direction and 1.1% in the y direction. The results confirm that the proposed model achieves high accuracy in capturing the large deformation behavior of spatial Timoshenko beams.

300

Table 1. Comparison of the Pre-bent beam tip displacements under a force applied at the free end

|                     | D      | Displacements (m) |       |     | Rel. Diff. (%) |     |  |  |
|---------------------|--------|-------------------|-------|-----|----------------|-----|--|--|
|                     | x      | y                 | z     | x   | y              | z   |  |  |
| Analytical Solution | -11.87 | -6.96             | 40.08 | -   | -              | -   |  |  |
| HAWC2               | -12.12 | -7.18             | 40.08 | 2.1 | 3.1            | 0.0 |  |  |
| Present             | -12.08 | -7.10             | 40.41 | 1.8 | 2.0            | 0.8 |  |  |

## 4.1.2 Large Deformation Analysis of a Thin Plate Beams under Concentrated Load

Fig. 5 illustrates a cantilevered thin plate beam with a total length of 0.51m, a cross-sectional width of 30mm, and a thickness of 1mm. The beam is made of 304 stainless steel, with a Young's modulus of 193 GPa and a Poisson's ratio of 0.3. To simulate concentrated loading, weights of 0.7N and 1.3N are suspended from the free end of the cantilever beam. For each case, the actual horizontal displacement u and vertical displacement v at three selected points along the beam are measured. In this

section, the proposed co-rotational beam model is employed to calculate the large deformation displacements of the cantilever beam under the two loading cases, and the plate beam is divided into 9 elements. Table 2 compares the results from the present study, the experimental measurements, and the data reported in Reference (Jiang et al., 2023)

Figure 5. Schematic Diagram of Thin Plate Beam

Table 2. Comparison of Nodal Displacements of the Thin Plates under Free-end Loading

| D 14                 |     | и            |                         |                     |              | ν                   |                       |                         |                      |              |                     |
|----------------------|-----|--------------|-------------------------|---------------------|--------------|---------------------|-----------------------|-------------------------|----------------------|--------------|---------------------|
| Case Positi on (x/L) | on  | Test<br>(mm) | ASU <sup>[5]</sup> (mm) | Rel.<br>Diff(<br>%) | Present (mm) | Rel.<br>Diff<br>(%) | Test<br>value<br>(mm) | ASU <sup>[5]</sup> (mm) | Rel.<br>Diff.<br>(%) | Present (mm) | Rel.<br>Diff<br>(%) |
|                      | 1/3 | -2.0         | -1.6                    | 20.0                | -1.9         | 5.0                 | -21.5                 | -23.5                   | 9.3                  | 22.6         | 5.1                 |
| I                    | 2/3 | -12.0        | -10.7                   | 10.8                | -11.3        | 5.3                 | -78.0                 | -80.9                   | 3.7                  | 77.8         | 0.3                 |
|                      | 1   | -24.0        | -26.0                   | 8.3                 | -26.5        | 10.4                | -150                  | -154.6                  | 3.1                  | 148.4        | 1.1                 |
|                      | 1/3 | -3.0         | -3.3                    | 10                  | 3.1          | 3.3                 | -28.0                 | -29.5                   | 5.4                  | 29.0         | 3.6                 |
| П                    | 2/3 | -17.0        | -19.1                   | 12.4                | 18.4         | 8.2                 | -96.0                 | -100.7                  | 4.9                  | 98.8         | 2.9                 |
|                      | 1   | -42.0        | -44.9                   | 6.9                 | 43.1         | 2.6                 | -185.0                | -190.6                  | 3.0                  | 187.1        | 1.1                 |

As shown in Table 2, except for the case 1, where the relative error of the horizontal displacement at the free end reached 10.4%, most of the other relative errors were within 10%, and this error remains stable as the applied load increases. In terms of the vertical displacement  $\nu$ , the proposed model produces the results with relative errors below 5%. Moreover, the predicted vertical deformation at the free end of the beam closely matches the measured values. These results validate the effectiveness and accuracy of the proposed model in capturing large elastic deformations in thin-walled flexible structures.

### 4.2Numerical analysis of variable cross-section beams

To validate the performance of the proposed model in handling variable geometric configurations, numerical simulations are conducted on two variable cross-section beams. The results obtained using the proposed model are compared with those from relevant literature to assess its accuracy and effectiveness.

327

330

335

340 341

### 4.2.1 Numerical analysis of a rectangular variable cross-section cantilever beam

The cantilever beam with a rectangular cross-section, as shown in Fig. 6, is considered. The beam has a total length of 10 m and a constant width of b = 0.25 m, its thickness tapers linearly from 1.0 m at the fixed end to 0.2 m at the free end. The elastic modulus of the material is  $E = 3.0 \times 10^4 GPa$ , and the beam is subjected to a concentrated vertical load of 10,000 N at the free end. To evaluate the performance of the model, the beam is discretized into 10 elements. The computed deflection and rotation at the free end are compared with the exact analytical solution and results from alternative method, as summarized in Table 3.

Figure 6. Simplified diagram of rectangular variable cross-section cantilever beam

As shown in Table 3, the deflection and rotation results obtained using the proposed model align exceptionally well with the analytical solution. The predicted deflection at the free end is 0.01530 m, exactly matching the analytical value, and the computed rotation is 0.00399 rad, with a relative difference of only 0.25%. In comparison, the segmental constant elements method yields a relative difference of 0.59% in deflection and 2.00% in rotation. This example demonstrates the effectiveness of the proposed co-rotational beam model in capturing the geometric nonlinear behavior of beams with varying cross-sections.

Table 3. Comparison of Deflection and Rotation at the Free End of a Rectangular Variable Cross-Section Cantilever Beam

|                                  | Deflection (m) | Rel. Diff. (%) | Rotation(rad) | Rel. Diff. (%) |
|----------------------------------|----------------|----------------|---------------|----------------|
| Analytical Solution              | 0.01530        | -              | 0.00400       | -              |
| Segmental Constant Elements [40] | 0.01521        | 0.59           | 0.00392       | 2.00           |
| Present                          | 0.01530        | 0.00           | 0.00399       | 0.25           |

### 4.2.2 Numerical simulation of a cantilever conical beam

A variable cross-section cantilever beam, as shown in Fig. 7, is analyzed with a total length of 10 m. At the free end, both the moment of inertia  $I_L$  and cross-sectional area  $A_L$  are one-third of those at the fixed end. The ratio of the beam length to the height of the cantilever end section is 50:1. The material properties include an elastic modulus of 210 GPa and a shear modulus

352

364

366

of 80.77 GPa. To facilitate comparison with existing numerical studies, a dimensionless load parameter  $\overline{\mathbf{F}} = FL^2/EI_L$ , as 347 defined in reference (Marjamäki and Mäkinen, 2009), is employed. 348

Fig. 8 shows the load-displacement response of the conical cantilever beam subjected to a vertically downward point load at its free end. The results from the proposed co-rotational beam model are compared with the numerical solution obtained using the Runge-Kutta method from reference (Marjamäki and Mäkinen, 2009) and with finite element results reported by Nguyen (Nguyen, 2013). As evident in Fig. 8, the response predicted by the proposed method aligns closely with the Runge-Kutta solution and shows improved agreement compared to Nguyen's finite element results. This comparison validates the accuracy and effectiveness of the proposed co-rotational model in capturing large deformation behavior in conical cantilever beams with variable cross-sections.

Figure 7. Conical beam

Nguyen2013 Runge Kutta Normalized displacements and rotation present work  $\bar{\theta}$  $\bar{v}$ 0.6  $|\bar{u}|$ 10 15 Normalized load

Figure 8. Normalized moment and deformation in tapered beam

### 4.3Variable taper frame model

A well-know benchmark frame structure (Manuel et al., 1968), shown in Fig. 9, is commonly used to assess the performance of nonlinear analysis methods. In its original configuration, members AB and BC possess constant stiffness. Building upon this example, Francisco(de Araujo et al., 2017) proposed a modified version by introducing variable stiffness to column AB, as illustrated in Fig. 10. The cross-sectional properties at points A and B for this modified configuration are provided as follows:

$$\begin{cases} I_{xA} = 14.76042 \times 10^{-8} m^4, I_{zA} = 17.04167 \times 10^{-8} m^4 \\ I_{xB} = 0.09375 \times 10^{-8} m^4, I_{zB} = 0.27083 \times 10^{-8} m^4 \\ S_A = 8.5 \times 10^{-4} m^2, S_B = 1.0 \times 10^{-4} m^2 \end{cases}$$

$$(46)$$

Beam BC retains a constant rectangular cross-section with  $S = 0.006 \, m^2$  and  $I = 2 \times 10^{-8} \, m^4$ . The material properties for the entire frame are assumed to be homogeneous, with an elastic modulus  $E = 7.2 \times 10^9 \, \text{GPa}$  and Poisson's ratio v = 0.3. The frame is discretized into 20 elements, and the interpolation method proposed in this study is applied. The resulting vertical and horizontal displacements at the load application points are compared with those obtained from Francisco2017 and a highly refined finite element mesh reported in reference (de Araujo et al., 2017). As shown in Fig. 11, the responses match quite well.

**Figure 9.** Frame scheme

**Figure 10.** Column geometry

Figure 11. Frame displacement at node 13

## 4.4Natural frequencies of the conical cantilever beam

This section considers an experimental conical cantilever beam reported in reference (Le et al., 2011) to verify the developed beam element. A modal analysis is performed where the natural frequencies are compared. The beam has a total length of 0.5*m*, with a fixed-end section diameter of 0.03*m*, and a free-end section diameter of 0.005*m*. The mass density and elastic modulus are 7800kg/m<sup>3</sup> and 210 GPa, respectively. The specific experimental setup is described in detail in reference (Le et al., 2011). The first five natural frequencies of the conical beam are computed using the proposed variable cross-section beam model and are compared with both experimental results and two numerical approaches from Ref. (Le et al., 2011). The comparison is presented in Table 4.

Table 4. Natural frequencies of the conical cantiliver beam

| Natural<br>modes | TMM using<br>Bessel functions<br>(errors) [Hz] % | TMM using cylindrical elements (errors) [Hz] % | Present result<br>(errors) [Hz] % | Experimental results [Hz] |
|------------------|--------------------------------------------------|------------------------------------------------|-----------------------------------|---------------------------|
| Model            | 160.7(1.1)                                       | 162.5(2.2)                                     | 166.4(4.6)                        | 159.0                     |
| Mode2            | 455.5(3.0)                                       | 457.4(3.4)                                     | 445.0(0.6)                        | 442.2                     |
| Mode3            | 962.8(7.3)                                       | 963.0(7.3)                                     | 920.2(2.5)                        | 897.5                     |
| Mode4            | 1702.0(6.9)                                      | 1699.0(6.7)                                    | 1658.9(4.2)                       | 1592.1                    |
| Mode5            | 2679.1(7.0)                                      | 2671.5(6.7)                                    | 2607.8(4.1)                       | 2504.0                    |

From Table 4, all three numerical methods produce results reasonably close to the experimental values. However, the proposed model demonstrates superior accuracy, with relative errors consistently below 5% across all five modes. In contrast, the relative errors of the TMM approaches in (Le et al., 2011) exceed 5% in several modes. Notably, the present model yields the most accurate results for the second and third modes, with relative errors of only 0.6% and 2.5%, respectively. These results confirm that the proposed variable cross-section beam model is effective in predicting the dynamic behavior of conical cantilever beams.

409

418

427

#### 5.Conclusions

In summary, a new approach for analyzing conical and geometrically nonlinear beam models has been successfully proposed by incorporating the analytical displacement shape functions and an improved co-rotational coordinate method to update the overall stiffness matrix. Firstly, the cantilever beam model with a constant cross-section and the cantilever thin plate experiment were compared with existing literature, and the obtained deformation results were found to be in close agreement with the actual results reported in the literature. Secondly, the cantilever beam and variable taper frame model considering geometric nonlinearity effects was examined, and the numerical calculations yielded deformation and rotation results that were consistent with the overall trends observed in the existing literature. Finally, the natural frequency of a conical beam was considered, and the numerical simulation results were compared with the simulated and experimental results from existing literature, demonstrating that the proposed model in this study exhibits higher computational accuracy. In this study, a new analytical approach for modeling conical and geometrically nonlinear beams is developed, incorporating analytical displacement shape functions and co-rotational formulation to update the global stiffness matrix. The proposed methodology is rigorously validated through four benchmark cases: (1) a pre-curved beam model, (2) large deformation analysis of thin plates, (3) conical beam modeling, and (4) variable-taper beam simulation. These validation studies systematically examine the method's capability to accurately model tapered beam configurations and precisely analyze geometrically nonlinear behavior in beams with varying cross-sections. Obtained results can be summarized as follows: Case 1: Validation studies were conducted by comparing the model predictions against established benchmark cases from literature. The constant cross-section cantilever beam analysis demonstrated excellent agreement with classical solutions, while the cantilever thin plate simulations closely matched experimental deformation patterns reported in previous studies. These comparisons confirmed the fundamental accuracy of the proposed formulation. Case 2: The nonlinear analysis capabilities were examined through simulations of large deformation scenarios. Both cantilever beam and variable taper frame configurations were investigated under geometrically nonlinear conditions. The obtained displacement and rotation results exhibited consistent trends with reference solutions, verifying the model's ability to handle nonlinear structural behavior. Case 3: Dynamic characteristics were evaluated through natural frequency analysis of conical beam structures. Comparative studies with existing numerical and experimental data revealed that the present model achieves superior accuracy in predicting vibrational behavior compared to conventional approaches. The improved performance is attributed to the precise representation of stiffness variations along the beam axis. In summary, this study develops and validates an advanced beam element formulation that successfully addresses two critical challenges in structural analysis: accurate modeling of variable cross-sections and robust simulation of geometric nonlinearity. The comprehensive validation framework demonstrates the method's reliability across multiple benchmark cases, establishing

## Acknowledgment

The authors are grateful for the support received from the National Natural Science Foundation of China [grant number 52078284, 52308318], Scientific Research Foundation of Inner Mongolia University (No.10000-23112101/056), Natural Science Foundation of Inner Mongolia Autonomous Region (No.22200-5233106), GuangDong Basic and Applied Basic Research Foundation [grant number 2022A0505020006, 2024A1515010090, STKJ2023067, 2022A1515010812]. Their financial support is gratefully acknowledged.

its potential for engineering applications requiring precise analysis of tapered beam structures under large deformations.

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
