# Peer review of "Geometric Nonlinear Analysis of Timoshenko Beams with Variable"

_Wind Energy Science, 2025_

## Referee Comment (RC5)

**Review Report**

**Manuscript Title:**

**Geometric Nonlinear Analysis of Timoshenko Beams with Variable Cross-Section Using Co-Rotational Formulation**

This paper addresses the geometrically nonlinear analysis of Timoshenko beam structures with variable cross-sections, which remains a challenging problem in computational structural mechanics. The authors propose a co-rotational finite element framework that incorporates an analytically derived displacement-based Timoshenko beam element for variable cross-sections, with the aim of improving both accuracy and computational efficiency in modeling large-deformation behavior.

The work is generally well developed and addresses a relevant research topic, with references that are consistent with the adopted methodology. Nonetheless, some concerns remain and should be adequately addressed.

- Although the modeling of Timoshenko co-rotational beams with tapered or variable cross-sections has been addressed in previous studies using multiple successful approaches, it remains unclear what constitutes the specific novelty or methodological advancement in the present work. Could the authors clarify how their formulation provides a substantive improvement or development over existing models?

- What is the practical advantage of incorporating analytically derived displacement shape functions within the co-rotational Timoshenko beam formulation, and how does this choice improve accuracy, computational efficiency, or overall performance compared to existing approaches?

- While the methodology presented in Section 2 attempts to account for geometric variability through Gaussian integration (Equation 16), there is a fundamental concern regarding the mathematical consistency of the local stiffness matrix derivation. The formulation in Equations (14 and 15) utilizes analytical shape functions originally developed for prismatic members;

however, applying these functions to non-prismatic elements without incorporating the spatial derivatives of the sectional properties (EI'($x$) and GA'($x$)) introduces a known field inconsistency. For elements with significant tapering, the omission of these gradient terms may lead to an inaccurate representation of the internal equilibrium, potentially affecting the overall robustness of this co-rotational framework.

- While the tangent stiffness formulation in Section 3 explicitly identifies the $K_m$ and $K^g$ matrices (as seen in Equation 42), the derivation lacks a detailed discussion on how these components are specifically adapted to the element's variable cross-section.

- Although the proposed formulation is applied to six numerical examples, the study does not provide a comparative assessment against existing methods for variable cross-section beams, which limits the demonstration of the approach's relative effectiveness and advantages.

- In Section 4, while the first two examples address 3D configurations, they are limited to constant cross-sections, whereas the subsequent four examples that account for variable sections are restricted to 2D analyses. The absence of a 3D example with a variable cross-section represents a significant gap, as it leaves the element's performance unverified in cases where spatial geometric nonlinearity and sectional tapering are coupled. Consequently, the authors must explicitly define the intended scope of this formulation and specify the categories of structural problems it is reliably applicable to, as the current results do not yet justify its robustness for complex, non-prismatic 3D applications.

- The abstract and introduction must explicitly define the research's novel contribution, clearly justifying the necessity of the proposed method instead of overemphasizing well-established classical theories

**Recommendation:**

Major revision is required.

---

## Referee Comment (RC6)

MS No.: wes-2025-180

Title: Geometric Nonlinear Analysis of Timoshenko Beams with Variable Cross-Section Using Co-rotational Formulation
Author(s): Xin Guo et al.

The authors formulate a co-rotational Timoshenko beam element for large displacement analysis of 3D frames, considering the influence of variable cross-section. The solution of the equilibrium equations is used to interpolate the lateral displacements and rotations for improving the element efficiency. Numerical examples and experiments are carried out to confirm the accuracy of the element. The topic is of interest, but many issues should be clarified for further evaluation of the paper.

- Primarily, the solution of the equilibrium equations of a Timoshenko beam with variable cross-section is difficult to derive. The author should provide more details on the derivation of the solution and the considered section profile. I believe that a solution for a general section cannot be derived. Additionally, the number of Gauss points used to evaluate the element tangent stiffness and mass matrices depends on the profile of the beam section. It is difficult to evaluate the element without knowing this information.

- The authors claim that "the proposed method achieves both high computational efficiency and accuracy in handling large deformations and nonlinear behavior", but the efficiency is not demonstrated in the paper.

- The presentation should be improved. Section 3 presents the co-rotational framework, which is well-known in the literature, but no references are cited. A 3D beam is considered, but the equilibrium equation (1) is written for a 2D beam. An explanation for Eqs. (1)-(3) should be given.

- More information on the large deformation behavior of the structure, such as snap-through and snap-back, is required to show the efficiency of the element and numerical algorithm.

---

## Author Comment (AC2)

**Reply to Reviewer #1's Comments**

This paper addresses the geometrically nonlinear analysis of Timoshenko beam structures with variable cross-sections, which remains a challenging problem in computational structural mechanics. The authors propose a co-rotational finite element framework that incorporates an analytically derived displacement-based Timoshenko beam element for variable cross-sections, with the aim of improving both accuracy and computational efficiency in modeling large-deformation behavior.

The work is generally well developed and addresses a relevant research topic, with references that are consistent with the adopted methodology. Nonetheless, some concerns remain and should be adequately addressed.

● Although the modeling of Timoshenko co-rotational beams with tapered or variable cross-sections has been addressed in previous studies using multiple successful approaches, it remains unclear what constitutes the specific novelty or methodological advancement in the present work. Could the authors clarify how their formulation provides a substantive improvement or development over existing models?

**Reply:**

Thank you for your comment. Our improvements to the large-deformation calculation method for variable-cross-section beams are primarily reflected in the following three aspects:

Improvement 1:

We propose a variable-cross-section Timoshenko beam element based on analytical displacement shape functions, which replaces traditional interpolation shape functions. This significantly enhances the computational accuracy of the element in geometrically nonlinear analysis.

Improvement 2:

Within the corotational framework, Gaussian integration is introduced to compute the stiffness and mass matrices of variable-cross-section beams. This avoids the repeated calculation of the moment of inertia for each cross-section required in conventional approaches, thereby improving computational efficiency. The method provides accurate and efficient solutions for beams with linearly varying width or thickness. For problems with pronounced taper—such as nonlinear variations in both width and thickness—additional section information is required to determine the additional unknown coefficients.

Improvement 3:

We propose a coordinate transformation method between local and global systems tailored for variable-cross-section beams. This approach can handle irregular sections and proportionally graded sections, extending the applicability of the corotational formulation.

● What is the practical advantage of incorporating analytically derived displacement shape functions within the co-rotational Timoshenko beam formulation, and how does this choice improve accuracy, computational efficiency, or overall performance compared to existing approaches?

**Reply:**

We thank the reviewer for the insightful comment. Compared with conventional interpolation-based shape functions, the analytically derived displacement shape functions—rigorously obtained from the equilibrium equations of the Timoshenko beam—offer the following three practical advantages:

(1)Improved Accuracy: Conventional interpolated shape functions introduce truncation errors, whereas the analytical shape functions can more accurately capture the actual bending deformation of the beam, thereby significantly enhancing computational accuracy in geometrically nonlinear analysis.

(2)Higher Computational Efficiency: The analytical shape functions can be directly incorporated into Gaussian quadrature, which reduces the required number of integration points and increases the efficiency in computing the element stiffness and mass matrices.

(3)Enhanced Numerical Stability: When dealing with problems involving large deformations and large rotations, the analytical shape functions better preserve the internal force equilibrium within the element, leading to improved overall numerical stability.

To further validate these points, we have added a new numerical example to demonstrate the reasonableness of the approach. The specific example is provided in the following reply.

● While the methodology presented in Section 2 attempts to account for geometric variability through Gaussian integration (Equation 16), there is a fundamental concern regarding the mathematical consistency of the local stiffness matrix derivation. The formulation in Equations(14 and 15) utilizes analytical shape functions originally developed for prismatic members; however, applying these functions to non-prismatic elements without incorporating the spatial derivatives of the sectional properties ($EI'(x)$ and $GA'(x)$) introduces a known field inconsistency. For elements with significant tapering, the omission of these gradient terms may lead to an inaccurate representation of the internal equilibrium, potentially affecting the overall robustness of this co-rotational framework.

**Reply:**

Thank you for the reviewer insightful comments. Following your suggestion, we have added explanatory text in the corresponding section of the paper. The details are provided below.

In the reference by Friedman and Kosmatka (1993), the variation of the moment of inertia and area of a non-uniform beam is linearly interpolated using a taper coefficient. Building on this, Nguyen (2013) further employs higher-order interpolation of area and moment of inertia based on the taper coefficient to calculate the deformation of tapered beams. In this paper, we reduce the number of unknowns by assuming linear variations in width and thickness. The corresponding unknown coefficients are determined using **Eq. 19** and then substituted into **Eq. 18** to compute the area and moment of inertia of the varying cross-sections. This approach provides relatively accurate results for beams with simple taper.

When addressing large deformations of beams with significant taper, however, higher-order interpolation of width and thickness becomes necessary. Each additional interpolation order introduces two more unknowns,

which means that additional known conditions — such as other cross-sectional dimensions or explicit

expressions for thickness and diameter along the beam length—are required. Only with such information can the shape function expressions accurately describe the large deformation behavior of beams with pronounced taper.

● While the tangent stiffness formulation in Section 3 explicitly identifies the Km and Kg matrices (as seen in Equation 42), the derivation lacks a detailed discussion on how these components are specifically adapted to the element's variable cross-section.

**Reply:**

We appreciate the reviewer's comment. The global stiffness matrix Kg in the overall coordinate system is composed of two parts: the linear stiffness Ka and the nonlinear stiffness Km. The linear stiffness Ka is formed from the element stiffness matrix Ke in the local coordinate system, and Ke inherently contains cross-section variation information. The nonlinear stiffness term is related to the internal forces at the current iteration step and the transformation matrix. The transformation matrix depends only on the degrees of freedom at the element's end nodes, which has the same effect for both variable and constant cross-sections. The internal force vector is related to the element stiffness matrix Ke in the local coordinate system and the nodal displacements, and this Ke also inherently contains cross-section variation information.

● Although the proposed formulation is applied to six numerical examples, the study does not provide a comparative assessment against existing methods for variable cross-section beams, which limits the

demonstration of the approach's relative effectiveness and advantages.

**Reply:**
Thank you for your comment. As per your suggestion, the corresponding section has been revised. This paper primarily compares the proposed calculation method for large deformation of variable-cross-section beams through three typical numerical examples (Examples 3–5), as detailed below:

**Example 3** examines a uniformly tapered variable-cross-section beam. The traditional method used in the references divides the beam into segments of uniform cross-sections for analysis. Compared to this approach, when the same number of elements is used, the results obtained by the method proposed in this paper are more accurate.

**Example 4** refers to the large deformation solution for variable-cross-section beams proposed by Nguyen et al., which combines the corotational formulation. In their approach, the cross-sectional area and moment of inertia are treated as linear functions of the taper coefficient. The results from this example show that the method proposed in this paper achieves higher computational accuracy compared to Nguyen's method.

**Example 5** involves the large deformation analysis of a non-uniformly tapered variable-cross-section frame. In the reference (Araujo et al., 2017), at the element level, a flexibility system of equations based on the principle of virtual forces (PVF) is established to calculate the tangent stiffness matrix and the equivalent nodal loads. The example results indicate that the method proposed in this paper achieves comparable computational accuracy to that of the referenced method.

● In Section 4, while the first two examples address 3D configurations, they are limited to constant cross-sections, whereas the subsequent four examples that account for variable sections are restricted to 2D analyses. The absence of a 3D example with a variable cross-section represents a significant gap, as it leaves the element's performance unverified in cases where spatial geometric nonlinearity and sectional tapering are coupled. Consequently, the authors must explicitly define the intended scope of this formulation and specify the categories of structural problems it is reliably applicable to, as the current results do not yet justify its robustness for complex, non-prismatic 3D applications.

**Reply:**
We thank the reviewer for the constructive comment. To verify the correctness of the proposed method for spatial deformation analysis of variable-cross-section beams and to demonstrate the robustness of the algorithm, the numerical example presented in the reference by Murín and Kutiš (2002) has been computed and validated. The corresponding results have been added to the **Section 4.5** in **Applications** of the paper.

**Figure 1** shows a 3D frame, with beams of varying circular cross-sections, loaded by concentrated loads F at node 1. The displacements of nodes 1 to 4 were founded. Variation of the cross-sectional area of the beams a is defined by the following diameter quadratic function $d(y) = 0.04 + 0.04y^2$. The beams b and c have constant diameters through lengths of elements. Detailed parameters can be found in (Murín et al. 2002). Only one exact beam element was used to model each beam (a, b, c). In the Hermite beam element model, only one element was used to represent the beams b and c in all cases, but beams a were modelled with 1, 2 and 3 elements in models 1, 2 and 3 respectively.

[Figure]

**Figure 1.** Frame displacement at node 13 (**Murín et al. 2002**)

The numerical results obtained by the present method are compared against those from the method proposed by Murín et al. (2002) and the solutions from classical Hermite beam elements, as presented in **Table 1**. It can be observed from the table that compared to the reference method, the displacement solutions of the present method at all nodes and under all loading cases are consistently closer to the exact solution, demonstrating a significant enhancement in computational accuracy. Furthermore, when the number of elements is varied, the present method exhibits a narrower and more stable variation range in its solutions, highlighting its superior numerical robustness.

**Table 1.** Comparison of results

| | Node1 (errors %) | | Node2 (errors %) | | Node3 (errors %) | | Node4 (errors %) | |
|---|---|---|---|---|---|---|---|---|
| | $U_x$(mm) | $U_z$(mm) | $U_x$(mm) | $U_z$(mm) | $U_x$(mm) | $U_z$(mm) | $U_x$(mm) | $U_z$(mm) |
| Exact solution | 0.775 | -1.098 | 0.774 | -0.428 | 0.945 | -0.428 | 0.945 | -1.098 |
| Model1 ref | 0.651(16.0) | -0.882(19.7) | 0.650 | -0.336(21.5) | 0.763(19.3) | -0.336 | 0.763 | -0.882 |
| Model1 this paper | 0.745(3.9) | -0.981(10.7) | 0.745 | -0.427(0.2) | 0.859(9.1) | -0.427 | 0.859 | -0.981 |
| Model2 ref | 0.743(4.1) | -1.008(8.2) | 0.722 | -0.390(8.9) | 0.869(8.0) | -0.390 | 0.869 | -1.008 |
| Model2 this paper | 0.767(1.0) | -1.085(1.2) | 0.766 | -0.423(1.2) | 0.933(1.3) | -0.423 | 0.933 | -1.086 |
| Model3 ref | 0.749(3.4) | -1.054(4.0) | 0.748 | -0.409(4.4) | 0.908(3.9) | -0.409 | 0.908 | -1.054 |
| Model3 this paper | 0.772(0.4) | -1.093(0.5) | 0.771 | -0.426(0.5) | 0.940(0.5) | -0.426 | 0.940 | -1.093 |

**Ref:**
[1] Murín, Justín, and Vladimír Kutiš. "3D-beam element with continuous variation of the cross-sectional area." Computers & structures 80.3-4 (2002): 329-338.

● The abstract and introduction must explicitly define the research's novel contribution, clearly justifying the necessity of the proposed method instead of overemphasizing well-established classical theories

**Reply:**
Thank you for your comment. The abstract, introduction, and conclusion sections of the manuscript have been revised accordingly to better highlight the contributions of this study.

---

## Author Comment (AC3)

**Reply to Reviewer #2's Comments**

The authors formulate a co-rotational Timoshenko beam element for large displacement analysis of 3D frames, considering the influence of variable cross-section. The solution of the equilibrium equations is used to interpolate the lateral displacements and rotations for improving the element efficiency. Numerical examples and experiments are carried out to confirm the accuracy of the element. The topic is of interest, but many issues should be clarified for further evaluation of the paper.

● Primarily, the solution of the equilibrium equations of a Timoshenko beam with variable cross-section is difficult to derive. The author should provide more details on the derivation of the solution and the considered section profile. I believe that a solution for a general section cannot be derived. Additionally, the number of Gauss points used to evaluate the element tangent stiffness and mass matrices depends on the profile of the beam section. It is difficult to evaluate the element without knowing this information.

**Reply:**

Thank you for the suggestions regarding these two key issues. In fact, Friedman, Zack, and J. B. Kosmatka et al. (1993) studied the solution method for the equilibrium equations of tapered Timoshenko beams. They performed linear interpolation for the variations of the beam's moment of inertia and cross-sectional area using a taper coefficient. For details, please refer to the paper (Le et al., 2011) exceeding "Exact stiffness matrix of a nonuniform beam—II. Bending of a Timoshenko beam." Computers & structures 49.3 (1993): 545–555. Building on this, Nguyen et al. (2013) calculated the deformation of tapered beams using higher-order interpolation of area and moment of inertia.

In this paper, the calculation of the cross-sectional variation follows two specific approaches:

(1) When only the moments of inertia and cross-sectional areas at certain sections are known. This paper reduces the number of undetermined coefficients by assuming linear variations in width and thickness, which is relatively accurate for simple tapered beams. However, when calculating large deformations of beams with significantly tapered cross-sections, higher-order interpolation for width and thickness is required. Each additional order introduces two more undetermined coefficients, meaning additional known conditions (such as cross-sectional areas and moments of inertia in the y and z directions at other sections) are necessary. Only then can the shape function expressions accurately represent large deformations of beams with significant taper.

(2) When the specific expressions for cross-sectional dimension variations (such as width or diameter) are known. This paper directly calculates the cross-sectional characteristics at the corresponding Gaussian integration points of the element using these expressions, followed by solution via Gaussian integration. In this case, the method achieves high accuracy and good robustness even for nonlinearly varying cross-sectional dimensions. Following your suggestion, relevant validation examples have been added in **Section 4.5 (Applications)** of the paper.

Regarding the number of Gaussian integration points depending on the variation of the beam cross-section, we have conducted extensive research. For example, in the five examples in the paper, we tested with 2, 4, 6, and 8 integration points, covering cases such as uniform cross-sections, uniformly tapered cross-sections, complex tapered cross-sections, and spatial deformations of tapered beams. The results showed little difference. Therefore, all examples in this paper were computed using four Gaussian integration points.

● The authors claim that "the proposed method achieves both high computational efficiency and accuracy in handling large deformations and nonlinear behavior", but the efficiency is not demonstrated in the paper.

**Reply:**

Thank you for your comment. This is indeed a crucial issue. The computational process within each element in the proposed method differs only in Timoshenko formulation of the relevant matrices, with no significant change in the computational efficiency of a single element. However, the proposed method can achieve relatively accurate results even with fewer element divisions, thereby improving the overall computational efficiency of

the algorithm to some extent. To validate this conclusion, a numerical example from the reference Murın, J., Justın, V., & Kutiš, V. (2002). "3D-beam element with continuous variation of the cross-sectional area." Computers & structures, 80(3-4), 329-338, was calculated and added to the Applications section of the paper. The results show that using only one element can achieve computational accuracy comparable to that obtained with three elements in the referenced study. The corresponding results have been added to the **Section 4.5** in **Applications** of the paper.

**Figure 1** shows a 3D frame, with beams of varying circular cross-sections, loaded by concentrated loads F at node 1. The displacements of nodes 1 to 4 were founded. Variation of the cross-sectional area of the beams a is defined by the following diameter quadratic function $d(y) = 0.04 + 0.04y^2$. The beams b and c have constant diameters through lengths of elements. Detailed parameters can be found in (Murín et al. 2002). Only one exact beam element was used to model each beam (a, b, c). In the Hermite beam element model, only one element was used to represent the beams b and c in all cases, but beams a were modelled with 1, 2 and 3 elements in models 1, 2 and 3 respectively.

[Figure]

**Figure 1.** Frame displacement at node 13 (**Murín et al. 2002**)

The numerical results obtained by the present method are compared against those from the method proposed by Murín et al. (2002) and the solutions from classical Hermite beam elements, as presented in **Table 1**. It can be observed from the table that compared to the reference method, the displacement solutions of the present method at all nodes and under all loading cases are consistently closer to the exact solution, demonstrating a significant enhancement in computational accuracy. Furthermore, when the number of elements is varied, the present method exhibits a narrower and more stable variation range in its solutions, highlighting its superior numerical robustness.

**Table 1.** Comparison of results

|  | Node1 (errors %) | | Node2 (errors %) | | Node3 (errors %) | | Node4 (errors %) | |
| --- | --- | --- | --- | --- | --- | --- | --- | --- |
|  | $U_x$(mm) | $U_z$(mm) | $U_x$(mm) | $U_z$(mm) | $U_x$(mm) | $U_z$(mm) | $U_x$(mm) | $U_z$(mm) |
| Exact solution | 0.775 | -1.098 | 0.774 | -0.428 | 0.945 | -0.428 | 0.945 | -1.098 |
| Model1 ref | 0.651(16.0) | -0.882(19.7) | 0.650 | -0.336(21.5) | 0.763(19.3) | -0.336 | 0.763 | -0.882 |
| Model1 this paper | 0.745(3.9) | -0.981(10.7) | 0.745 | -0.427(0.2) | 0.859(9.1) | -0.427 | 0.859 | -0.981 |
| Model2 ref | 0.743(4.1) | -1.008(8.2) | 0.722 | -0.390(8.9) | 0.869(8.0) | -0.390 | 0.869 | -1.008 |
| Model2 this paper | 0.767(1.0) | -1.085(1.2) | 0.766 | -0.423(1.2) | 0.933(1.3) | -0.423 | 0.933 | -1.086 |
| Model3 ref | 0.749(3.4) | -1.054(4.0) | 0.748 | -0.409(4.4) | 0.908(3.9) | -0.409 | 0.908 | -1.054 |
| Model3 this paper | 0.772(0.4) | -1.093(0.5) | 0.771 | -0.426(0.5) | 0.940(0.5) | -0.426 | 0.940 | -1.093 |

● The presentation should be improved. Section 3 presents the co-rotational framework, which is well-known in the literature, but no references are cited. A 3D beam is considered, but the equilibrium equation (1) is written for a 2D beam. An explanation for Eqs. (1)-(3) should be given.

**Reply:**

Thank you for your valuable suggestions. Citations have been added to the co-rotational formulation derivation section in the manuscript as recommended, with the corresponding references provided below. Regarding the three-dimensional beam problem, the expressions for the y-direction and z-direction in Equations (1-3) are analogous. A supplementary explanation has been incorporated into the relevant section of the main text.

**Ref:**

[1] Crisfield MA.: A consistent co-rotational formulation for non-linear, three-dimensional, beam-elements[J]. Comput Method Appl M 1990;81(2):131-150.

[2] Crisfield MA, Moita GF.: A unified co-rotational framework for solids, shells and beams[J]. Int J Solids Struct 1996;33(20-22):2969-2992.

● More information on the large deformation behavior of the structure, such as snap-through and snap-back, is required to show the efficiency of the element and numerical algorithm.

**Reply:**

Thank you for your suggestion. The issue you raised is of significant research value. Analyzing complex post-buckling behaviors involving snap-through and snap-back indeed serves as a rigorous benchmark for evaluating the robustness of nonlinear beam elements and algorithms.

In this paper, our primary objective is to develop an efficient co-rotational formulation for three-dimensional variable cross-section Timoshenko beams and to validate its accuracy and convergence on a series of fundamental yet critical large-displacement and large-rotation problems. We believe these results establish a reliable foundation for the proposed method.

The type of strongly nonlinear problems involving limit points and unstable paths that you highlighted indeed represents a very important and natural extension of this method's application. As demonstrated in the reference Battini, J.-M. (2008), similar problems have also been analyzed using the co-rotational method. We also plan to dedicate future work to specifically investigate and demonstrate the application of this method for full-path tracking of post-buckling behavior in structures such as arches and domes. Once again, we appreciate this constructive suggestion, which has helped clarify the key directions for our future research.

**Ref:**

[3] Battini, Jean-Marc. "A non-linear corotational 4-node plane element." Mechanics research communications 35.6 (2008): 408-413.

---

## Author Comment (AC4)

**Geometric Nonlinear Analysis of Timoshenko Beams with Variable Cross-Section Using Co-rotational Formulation**

Xin Guo[1,2,3], Hailiang Feng[1], Jiajun Hou[1], Yanpei Gao[4], Dongsheng Li[4], Peng Guo[4]

[1]Department of Bridge Engineering, School of Transportation Institute, Inner Mongolia University, Hohhot 010070, China

[2]Inner Mongolia University Structure Testing Key Laboratory, Inner Mongolia University, Hohhot 010070, China;

[3]Inner Mongolia Engineering Research Center of Testing and Strengthening for Bridges, Inner Mongolia University, Hohhot 010070, China;

[4]MOE Key Laboratory of Intelligent Manufacturing Technology, Department of Civil and Environmental Engineering, Guangdong Engineering Center for Structure Safety and Health Monitoring, Shantou University, Shantou Key Laboratory of Offshore Wind Energy, Shantou 515063, China

*Correspondence to*: Yanpei Gao(744962238@qq.com), Dongsheng Li( lids@stu.edu.cn)

**Abstract.** The geometrically nonlinear analysis of Timoshenko beams with variable cross-sections remains a challenging task in engineering practice, particularly for structures subjected to large deformations. While co-rotational (CR) formulations have been widely adopted for geometric nonlinear analysis, most existing CR-based beam models assume constant cross-sectional properties, limiting their applicability to beams with variable geometries. To overcome this limitation, this study introduces a novel co-rotational formulation specifically tailored for variable cross-section Timoshenko beams. The proposed approach integrates two key innovations: (1) the development of an improved spatial Timoshenko beam element employing analytical displacement shape functions to accurately capture bending deformation in variable cross-sections, and (2) the introduction of an efficient Gaussian integration scheme for computing stiffness and mass matrices, eliminating the need for explicit moment-of-inertia evaluations at each cross-section. The tangent stiffness matrix is systematically derived within the co-rotational framework. The method is validated through five benchmark examples, including comparisons with experimental data and numerical results from the literature. Results demonstrate that the proposed model achieves superior computational accuracy and efficiency in handling large deformations, dynamic responses, and nonlinear behaviors of beams with irregular or proportionally graded cross-sections, offering a robust alternative to existing variable cross-section beam formulations.

[revised manuscript text omitted]

Nevertheless, most existing CR formulations assume uniform cross-sectional properties, which significantly restricts their applicability to modern designs employing tapered or functionally graded beams. Although a few studies have attempted to incorporate cross-sectional variations within the CR framework, they often suffer from inadequate accuracy or computational inefficiency, especially when the cross-section changes abruptly or the beam undergoes large rotations.

To overcome these persisting challenges, this paper presents a refined co-rotational beam model specifically designed for variable cross-sections, with three principal contributions:

A novel variable-cross-section Timoshenko beam element is formulated using analytical displacement shape functions derived from the equilibrium equations of a Timoshenko beam. This approach eliminates the truncation errors associated with standard

polynomial interpolations and provides a more accurate description of the bending deformation, thereby enhancing the overall precision of the co-rotational procedure.

An efficient numerical integration strategy based on Gaussian quadrature is introduced to compute the element stiffness and mass matrices. This strategy avoids the need to explicitly evaluate the moment of inertia at each cross-section, leading to a substantial reduction in computational cost while maintaining accuracy.

A consistent tangent stiffness matrix is derived within the co-rotational framework, explicitly accounting for the geometric nonlinearities induced by large displacements and rotations. The formulation is general enough to accommodate both irregular and proportionally tapered cross-sections, extending the applicability of CR methods to a broader class of engineering structures.

The remainder of this paper is structured as follows: Section 2 develops the improved stiffness and mass matrices for the variable cross-section beam element. Section 3 describes the co-rotational formulation for geometric nonlinear analysis. Section 4 validates the proposed model through a series of benchmark examples, including constant and variable cross-section beams, a tapered frame, and a dynamic frequency analysis. Finally, the main conclusions of this investigation are thereafter summarized in Section 5.

**2.The improved spatial Timoshenko beam element with variable cross-section**

The CR method enables the use of linear Timoshenko beam elements to derive the tangent stiffness matrix in the global coordinate system. Typically, interpolated shape functions are employed to construct the beam element. However, most of these shape functions approximate beam displacements, which introduces truncation errors and decreases computational accuracy. In this section, an improved Timoshenko beam element with a variable cross-section is proposed to improve computational accuracy by employing analytical displacement shape functions for bending deformation. The specific process is outlined below.

As illustrated in Fig. 1, a beam with variable cross-section is considered. The beam element has a total length $L$, with the coordinate origin at the left end. The $x$-axis is aligned with the longitudinal direction, while the $y$- and $z$- axes align with the principal axis of the cross-section. Typically, the displacement at any point within the spatial beam element is represented by $\{u, v, w, \theta_x, \theta_y, \theta_z\}$. where $u$ is the axial displacement along the $x$ axis, $v$ and $w$ are the transverse displacements along the $y$ and $z$ axis, respectively, and $\theta_x, \theta_y, \theta_z$ denote the rotations about the $x$, $y$, and $z$ axis, respectively. The cross-section parameters are defined: where $b$ is the width, $h$ is the thickness, $S$ is the cross-sectional area, $I_y$ and $I_z$ are the moments of inertia about the $y$- and $z$-axis, respectively.

[Figure]

**Figure 1.** Variable geometric properties in a tapered beam

Define $k_y$ and $k_z$ as the cross-sectional non-uniformity coefficients along the $y$ and $z$ axes, respectively, $E$ as the elastic

modulus, $G$ as the shear modulus, and $J$ as the moment of inertia. Substituting the constitutive relations and geometric equations of the Timoshenko beam into the equilibrium equations yields:

$$\begin{cases} \frac{\partial}{\partial x}\left[EI_y \frac{\partial \theta_y}{\partial x}\right] = k_z GA\left(\frac{\partial w}{\partial x} + \theta_y\right) \\ k_z GA\left(\frac{\partial^2 w}{\partial x^2} + \frac{\partial \theta_y}{\partial x}\right) = 0 \end{cases}, \tag{1}$$

For clarity and conciseness in presentation, the equilibrium equations are initially presented in the x-z plane (2D form). The formulation in the x-y plane is analogous, following the same principle by substituting corresponding variables (e.g., replacing $w$ with $v$, $\theta_y$ with $\theta_z$, $I_y$ with $I_z$, and $k_z$ with $k_y$). This approach does not compromise generality, as the two bending directions are decoupled within the linear local element formulation.

The relationship between transverse displacement and bending displacement is given by:

$$w = w_b - \frac{EI_y}{k_z GA}\frac{\partial^2 w_b}{\partial x^2} + \frac{EI_y}{k_z GA}\frac{\partial^2 w_b}{\partial x^2}|x = 0, \tag{2}$$

where subscripts b denoting contributions from bending deformation respectively.

Similarly, the analytical solution of transverse displacement $v$ satisfying the boundary conditions can be obtained as:

$$v = v_b - \frac{EI_z}{k_y GA}\frac{\partial^2 v_b}{\partial x^2} + \frac{EI_z}{k_y GA}\frac{\partial^2 v_b}{\partial x^2}|x = 0, \tag{3}$$

Similar to the traditional Timoshenko beam element, the displacements in the $u$ and $\theta_x$ are interpolated linearly. While the transverse displacements $v_b$ and $w_b$ are interpolated using cubic polynomial, and their expressions are given by:

$$\begin{cases} u(x) = c_1 x + c_2 \\ \theta_x(x) = c_{11}x + c_{12} \\ v_b(x) = c_3 x^3 + c_4 x^2 + c_5 x + c_6 \\ w_b(x) = c_7 x^3 + c_8 x^2 + c_9 x + c_{10} \end{cases}, \tag{4}$$

In general, the strain vector of a spatial Timoshenko beam element is expressed as:

$$\begin{cases} \boldsymbol{\varepsilon} = \left[\varepsilon_x, \gamma_y, \gamma_z, \gamma_x, \varepsilon_y, \varepsilon_z\right]^T \\ = \left[\frac{\partial u}{\partial x}, \frac{\partial v}{\partial x} - \theta_z, \frac{\partial w}{\partial x} + \theta_y, \frac{\partial \theta_x}{\partial x}, \frac{\partial \theta_y}{\partial x}, \frac{\partial \theta_z}{\partial x}\right]^T \\ = \boldsymbol{\varepsilon}_\alpha + \boldsymbol{\varepsilon}_\beta \end{cases}$$

$$, \tag{5}$$

where $\boldsymbol{\varepsilon}_\alpha = \left[\frac{\partial u}{\partial x}, \frac{\partial v}{\partial x}, \frac{\partial w}{\partial x}, \frac{\partial \theta_x}{\partial x}, \frac{\partial \theta_y}{\partial x}, \frac{\partial \theta_z}{\partial x}\right]^T$ and $\boldsymbol{\varepsilon}_\beta = \left[0, -\theta_z, \theta_y, 0,0,0\right]^T$. By combining Eqs. (4) and (5), the expressions for the displacement and rotation vector $\boldsymbol{u}(x)$ of the beam can be obtained as follows:

$$\boldsymbol{u}(x) = \boldsymbol{A}(x)\boldsymbol{c}, \tag{6}$$

where the matrix $\boldsymbol{A}(x)$ represents the displacement-rotation coefficient matrix with respect to the shape function coefficient vector $\boldsymbol{c} = \{c_1, \cdots, c_{12}\}^T$. Taking the derivative of Eq. (6) yields:

$$\begin{cases} d\boldsymbol{u}(x) = \left\{\frac{\partial u}{\partial x}, \frac{\partial v}{\partial x}, \frac{\partial w}{\partial x}, \frac{\partial \theta_x}{\partial x}, \frac{\partial \theta_y}{\partial x}, \frac{\partial \theta_z}{\partial x}\right\}^T \\ d\boldsymbol{u}(x) = d\boldsymbol{A}(x)\boldsymbol{c} \end{cases}, \tag{7}$$

Based on the boundary conditions at $x=0$ and $x=L$, the relationship between the shape function coefficients and the nodal displacements can be derived and expressed in matrix form as follows:

$$\boldsymbol{H}(x)\boldsymbol{c} = \boldsymbol{d}, \tag{8}$$

where $\boldsymbol{H}(x)$ is the coefficient matrix of the shape function coefficients. The nodal displacement $\boldsymbol{d}$ is expressed as:

$$\boldsymbol{d} = \{u_1, v_1, w_1, \theta_{x1}, \theta_{y1}, \theta_{z1}, u_2, v_2, w_2, \theta_{x2}, \theta_{y2}, \theta_{z2}\}^T, \tag{9}$$

By substituting Eq. (9) into Eqs. (6) and (7), the following expressions are obtained:

$$u(x) = A(x)H(x)^{-1}d, \tag{10}$$

$$du(x) = dA(x)H(x)^{-1}d, \qquad\qquad , \tag{11}$$

The relationship between the strain and nodal displacements of the element is then given by:

$$\begin{cases} \varepsilon_\alpha = du(x) = dA(x)H(x)^{-1}d \\ \varepsilon_\beta = T_N u(x) = T_N A(x)H(x)^{-1}d \\ \varepsilon = \varepsilon_\alpha + \varepsilon_\beta = [dN(x) + T_N N(x)]d = B(x)d \end{cases} , \tag{12}$$

where $N(x) = A(x)H(x)^{-1}$, $dN(x) = dA(x)H(x)^{-1}$, $B(x)$ is the strain-displacement matrix, and $T_N$ satisfies the following relationship:

$$T_N = \begin{bmatrix} 0 & 0 & 0 & 0 & 0 & 0 \\ 0 & 0 & 0 & 0 & 0 & -1 \\ 0 & 0 & 0 & 0 & 1 & 0 \\ 0 & 0 & 0 & 0 & 0 & 0 \\ 0 & 0 & 0 & 0 & 0 & 0 \\ 0 & 0 & 0 & 0 & 0 & 0 \end{bmatrix}, \tag{13}$$

By numerically integrating over the length $L$ of the beam, the element stiffness matrix $K_e$ and mass matrix $M_e$ of the variable cross-section Timoshenko beam element are formulated as:

$$\begin{cases} K_e = \int_0^L B(x)^T K_{cs}(x)B(x)dx \\ M_e = \int_0^L N(x)^T M_{cs}(x)N(x)dx \end{cases} , \tag{14}$$

Define J as the moment of inertia. The sectional stiffness matrix $K_{cs}(x)$ for a variable cross-section beam is expressed as:

$$K_{cs}(x) = diag[ES(x), k_y GS(x), k_z GS(x), GJ(x), EI_y(x), EI_z(x)], \tag{15}$$

Directly evaluating the integrals in Eq. (14) for variable cross-sections is often computationally intensive. Therefore, in this study, Gaussian quadrature is introduced to efficiently compute the element stiffness and mass matrices of the variable cross-section beam:

$$\begin{cases} K_e = \sum_{i=1}^{n} \frac{L}{2}\omega_i B(x_i)^T K_{cs}(x_i)B(x_i) \\ M_e = \sum_{i=1}^{n} \frac{L}{2}\omega_i N(x_i)^T M_{cs}(x_i)N(x_i) \end{cases} , \tag{16}$$

where $n$ is the number of Gaussian integration points, $\omega_i$ and $x_i$ are the corresponding weight coefficients and integration nodes, respectively.

The stiffness and mass matrices of the cross-section are determined based on the relevant parameters of the cross-section. Considering the diverse forms of cross-sections, a general formula is provided here to handle the cross-sectional parameters of variable cross-section beams with a certain taper.

Assuming that the aspect ratio of the variable cross-section beam remains constant, i.e.

$$\frac{b_r}{b_l} = \frac{h_r}{h_l}, \tag{17}$$

where, $b_r$ and $h_r$ are the width and thickness of the cross-section at the right end, and $b_l$ and $h_l$ are the width and thickness at the left end. Under this assumption, the cross-sectional parameters at any arbitrary point along the beam can be expressed as:

$$\begin{cases} h(x) = k_1 x + f_1 \\ b(x) = k_2 h(x) = k_2(k_1 x + f_1) \\ S(x) = p_1 b(x)h(x) = p_1 k_2(k_1 x + f_1)^2 = k_3(k_1 x + f_1)^2 \\ I_1(x) = p_2 b(x)h(x)^3 = p_2 k_2(k_1 x + f_1)^4 = k_4(k_1 x + f_1)^4 \end{cases} , \tag{18}$$

The calculation of the cross-sectional parameters for each cross-section requires solving for the corresponding coefficients $f_l$ and $k_i(i = 1,3,4)$. The transition coefficients $k_2$, $p_1$ and $p_2$ do not need to be solved. This can be achieved by solving using the relevant parameters of the cross-section at both ends of the beam. For the fixed end of the beam, when $x=0$, we have $h =$

$h_l$, $S = S_{max}$, $I_y = I_{ymax}$. when $x=L$, we have $S = S_{min}$, $I_y = I_{ymin}$. By substituting the known parameters of the beam at both ends into Eq. (18), we obtain:

$$k_1 = h_1 \left( \sqrt{\frac{S_{min}}{S_{max}}} \right) / L$$

$$k_3 = S_{max}/h_1^2 \tag{19}$$

$$k_4 = I_{y\ max}/h_1^4$$

When the aspect ratio of the structure is variable, the width and thickness of the cross-section are mutually independent, By measuring the maximum thicknesses $h_{lmax}$ and $h_{rmax}$ of the cross-sections perpendicular to the y-axis at both ends of the structure, the expression for the cross-sectional parameters at any point within the unit can also be derived.

The variation pattern of cross-sections is classified into two categories:

(1) Only partial cross-sectional moments of inertia and cross-sectional areas are known. In this paper, by assuming linear variation of width and thickness, the number of undetermined coefficients is reduced, which proves to be relatively accurate for the calculation of simple tapered beams. When calculating large deformations of beams with significantly tapered cross-sectional variations, higher-order interpolation is required for width and thickness. Each additional order introduces two additional undetermined coefficients, necessitating extra known conditions (such as cross-sectional areas and moments of inertia in the y- and z-directions at other sections). Only under these conditions can the derived shape function expressions accurately represent the large deformations of beams with notably tapered cross-sectional variations.

(2) The specific expression for the variation of cross-sectional dimensions (such as width or diameter) is known. In this paper, the cross-sectional characteristics at the Gaussian integration points of the element can be directly computed using the expression for dimensional variation, and the solution is then obtained through Gaussian integration. Under such circumstances, this method demonstrates high accuracy and strong robustness even for nonlinearly varying cross-sectional dimensions.

Once the relevant coefficients are obtained, they can be substituted into the coordinates of the Gaussian integration points to calculate the cross-sectional parameters. By substituting the cross-sectional parameters into Eqs. (14) and (15), the element stiffness matrix $K_e$ and the element mass matrix $M_e$ of the variable cross-section Timoshenko beam element can be obtained.

**3.Co-rotational formulation**

The co-rotational formulation stands out by extracting the elastic deformation displacements from the overall displacements (Crisfield, 1990; Crisfield and Moita, 1996; Crisfield et al., 1997), thus predefining the projection relationship. The motion of the beam element from its initial state to the final deformed state is decomposed into rigid body motion and pure deformation. The rigid body motion component encompasses the rigid translation and rotation in the local reference coordinate system. Therefore, the core challenge of the co-rotational formulation lies in handling the coordinate transformation between different frames, thereby establishing the relationship between pure deformation and the overall deformation.

**3.1Definition and transformation of the reference coordinate system for spatial beam elements**

For the spatial two-node beam element, the reference coordinate system is defined as shown in Fig. 2. The unit orthogonal vectors $E_i, i = 1,2,3$, represent the global reference system of the beam element, which remains fixed and unchanged. The unit orthogonal vectors $E_i^h, i = 1,2,3$, represent the local reference system of the beam element after rigid body motion, which continuously translates and rotates with the beam element. The local reference system $E_i^q, i = 1,2,3$ represents the original coordinate system of the beam element before deformation. Additionally, the vectors $e_i^1$ and $e_i^2$, define the cross-sectional reference system of the two nodes (1 and 2) of the beam.

[Figure]

**Figure 2.** Beam kinematics and coordinate systems

First, the rigid rotation of the local coordinate system $\boldsymbol{E}_i^h$ is addressed. The rigid rotation matrix $\boldsymbol{R}_r$ represents the transformation matrix from the reference system $\boldsymbol{E}_i$ to $\boldsymbol{E}_i^h$, and its expression is given by:

$$\boldsymbol{R}_r = [r_1 \quad r_2 \quad r_3] \tag{20}$$

The vector $\boldsymbol{r}_1$ is computed as the line connecting node 1 and node 2 of the beam element before and after deformation:

$$\boldsymbol{r}_1 = \frac{s_2^g - s_1^g}{l}, \tag{21}$$

where $s_i^g$ represents the coordinates of node $i$ in the global reference system after rigid rotation. The length $l$ of the beam after deformation can be obtained by $l = \left\| s_2^g - s_1^g \right\|$.

The directions of the remaining two axes are determined by introducing an auxiliary vector $\boldsymbol{q}$. The auxiliary vector serves two main purposes: (1) to solve the rigid rotation matrix in the global coordinate system; (2) to determine the differential relationship between the rigid rotation angle and the total displacement of the structure. Initially, the direction of $\boldsymbol{q}$ aligns with the local coordinate axis $\boldsymbol{E}_2^q$. After deformation of the beam element, the determination of the auxiliary vector $\boldsymbol{q}$ is related to the transformation of the local reference system:

$$\boldsymbol{q}_i = \boldsymbol{R}_i^g \boldsymbol{R}_0 [0 \quad 1 \quad 0]^{\mathrm{T}}, i = 1,2, \tag{22}$$

$$\boldsymbol{q} = \frac{1}{2}(\boldsymbol{q}_1 + \boldsymbol{q}_2), \tag{23}$$

where $\boldsymbol{R}_1^g$ and $\boldsymbol{R}_2^g$ are the orthogonal matrices corresponding to the directions of the end nodes $\boldsymbol{e}_i^1$ and $\boldsymbol{e}_i^2$, respectively. $\boldsymbol{q}_1$ and $\boldsymbol{q}_2$ are the directions of the left and right end reference systems of the local reference system $\boldsymbol{E}_2^q$ after rigid rotation. $\boldsymbol{R}_0$ denotes the initial orientation of the local coordinates, and $\boldsymbol{q}$ represents the direction of the local reference system $\boldsymbol{E}_2^q$ after rigid rotation.

By combining Eqs. (21), (22), and (23), the expressions for the remaining two components of the orthogonal matrix $\boldsymbol{R}_r$ can be obtained:

$$\boldsymbol{r}_3 = \frac{r_1 \times q}{\|r_1 \times q\|} \quad \boldsymbol{r}_2 = \boldsymbol{r}_3 \times \boldsymbol{r}_1, \tag{24}$$

The local rotation matrix of the coordinate axis is defined as $\bar{R}_i$, and the transformation from $E_i$ to $e_i^1$ and $e_i^2$ can be expressed as follows:

$$R_r\bar{R}_i = R_i^g R_0, i = 1,2, \tag{25}$$

Since $R_r^T R_r = I$, Eq. (25) can be transformed as follows:

$$\bar{R}_i = R_r^T R_i^g R_0, i = 1,2, \tag{26}$$

Thus, the local rotation angles can be obtained as follows:

$$\bar{\vartheta}_i = log(\bar{R}_i), \tag{27}$$

**3.2 Transformation of displacement vectors between the local and global coordinate systems**

The global displacement vector of the beam element is defined as $P_g^g$, and the displacement vector in the local coordinate system after removing rigid body deformations is denoted as $P_l$. By utilizing the rotation framework described in the previous section, the local displacement $P_l$ is obtained by subtracting the rigid body displacement from the total displacement $P_g^g$. The local internal force vector $f_l$ and the tangent stiffness matrix $K_l$ in the local coordinate system are computed through the transformation relationship between the two. The expression of the internal force vector $F_g$ in the global coordinate system can be derived by balancing the internal virtual work in the global and local systems:

$$V = \delta P_l^T f_l = \delta P_g^{gT} F_g, \tag{28}$$

The variations of the displacement vectors $P_g^g$ and $P_l$ can be expressed as follows:

$$\delta P_l = [\delta\bar{u} \quad \delta\bar{\vartheta}_1^T \quad \delta\bar{\vartheta}_2^T]^T, \tag{29}$$

$$\delta P_g^g = [\delta u_1^{gT} \quad \delta\theta_1^{gT} \quad \delta u_2^{gT} \quad \delta\theta_2^{gT}]^T, \tag{30}$$

where, $\delta\bar{\vartheta}_i, (i = 1,2)$ represents the variation of spatial rotation angles in the local coordinate system after considering rigid body deformations, and $\partial\theta_i^g (i = 1,2)$ represents the variation of spatial rotation angles in the global coordinate system.

The variation of the transformation matrix involves the formation of a new matrix composed of rotational angles:

$$\delta\bar{R}_i = \delta\widetilde{\bar{\theta}}_i \bar{R}_i, \tag{31}$$

where the superscript tilde denotes the skew-symmetric matrix corresponding to a vector. A new local coordinate system, denoted as $P_a$, is defined based on Eqs. (29) and (31).

$$P_a = [\bar{u}^T \quad \bar{\theta}_1^T \quad \bar{\theta}_2^T]^T, \tag{32}$$

Let $f_a$ represents the internal force vector corresponding to $\delta P_a$, and $K_l$ denotes the transformed local stiffness matrix $K_e$ obtained in Section 2 of this paper, which is converted to a 7-degree-of-freedom system. The transformation matrix between vectors $P_a$ and $P_l$ can be obtained through the transformation relationship of their respective stiffness matrices. The final conversion of $K_l$ to $K_a$ can be expressed as follows:

$$K_a = B_l^T K_l B_l + K_h, K_h = \begin{bmatrix} 0 & 0_{1\times3} & 0_{1\times3} \\ 0_{3\times1} & K_{h1} & 0_{3\times3} \\ 0_{3\times1} & 0_{3\times3} & K_{h2} \end{bmatrix}, \tag{33}$$

The matrix $B_l$ can be directly obtained by rotating the vector. The expressions for $K_{h1}$ and $K_{h2}$ are derived from the following equation:

$$\frac{\partial}{\partial\theta}[T_s^{-T}v] = \frac{\partial}{\partial\bar{\vartheta}}[T_s^{-T}v]\frac{\partial\bar{\vartheta}}{\partial\theta} = \frac{\partial}{\partial\bar{\vartheta}}[T_s^{-T}v]T_s^{-1}, \tag{34}$$

$$T_s(\Phi) = \frac{sin\varphi}{\varphi}I + (1 - \frac{sin\varphi}{\varphi})ee^T + \frac{1}{2}(\frac{sin(\varphi/2)}{\varphi/2})^2\widetilde{\Phi}, \tag{35}$$

where $v$ represents the bending moment acting on the two ends of the internal force vector in the local coordinate system, $e$ is the unit vector corresponding to the angle vector, $K_{h1}$ and $K_{h2}$ correspond to $\bar{\vartheta}_1$ and $\bar{\vartheta}_2$ in Eq. (34). Consequently, the differential relationship between the rotational vector in the local coordinate system and the displacement vector in the global

coordinate system can be derived as follows:

$$\begin{bmatrix} \delta\bar{\boldsymbol{\theta}}_1 \\ \delta\bar{\boldsymbol{\theta}}_2 \end{bmatrix} = \left( \begin{bmatrix} 0 & \boldsymbol{I} & 0 & 0 \\ 0 & 0 & 0 & \boldsymbol{I} \end{bmatrix} - \begin{bmatrix} \boldsymbol{G}_\theta{}^{\mathrm{T}} \\ \boldsymbol{G}_\theta{}^{\mathrm{T}} \end{bmatrix} \right) \boldsymbol{E}^{\mathrm{T}}\delta\boldsymbol{P}_g^g = \boldsymbol{P}\boldsymbol{E}^{\mathrm{T}}\delta\boldsymbol{P}_g^g, \tag{36}$$

where $\boldsymbol{G}_\theta = \frac{\partial\boldsymbol{\theta}_r^e}{\partial\boldsymbol{P}_g^g}, \boldsymbol{E} = diag[\boldsymbol{R}_r \quad \boldsymbol{R}_r \quad \boldsymbol{R}_r \quad \boldsymbol{R}_r]$.

Thus, the relationship between $\delta\boldsymbol{P}_a$ and $\partial\boldsymbol{P}_g^g$ can be obtained as follows:

$$\delta\boldsymbol{P}_a = \boldsymbol{B}_a\delta\boldsymbol{P}_g^g, B_a = \begin{bmatrix} \boldsymbol{r} \\ \boldsymbol{P}\boldsymbol{E}^{\mathrm{T}} \end{bmatrix}, \tag{37}$$

where $\boldsymbol{r} = [-\boldsymbol{r}_1^{\mathrm{T}} \quad \boldsymbol{0}_{1\times 3} \quad \boldsymbol{r}_1^{\mathrm{T}} \quad \boldsymbol{0}_{1\times 3}]$. The matrix $\boldsymbol{G}_\theta$ in Eq. (36) is related to $\delta\boldsymbol{\theta}_r^e$.

$$\delta\tilde{\boldsymbol{\theta}}_r^e = \boldsymbol{R}_r^{\mathrm{T}}\delta\boldsymbol{R}_r, \delta\boldsymbol{\theta}_r^e = \begin{bmatrix} -\boldsymbol{r}_2^{\mathrm{T}}\delta\boldsymbol{r}_3 \\ -\boldsymbol{r}_3^{\mathrm{T}}\delta\boldsymbol{r}_1 \\ \boldsymbol{r}_2^{\mathrm{T}}\delta\boldsymbol{r}_1 \end{bmatrix}, \tag{38}$$

The expression for $\boldsymbol{r}_1, \boldsymbol{r}_2, \boldsymbol{r}_3$, and $\delta\boldsymbol{r}_1$ can be easily obtained. As for $\delta\boldsymbol{r}_3$, it is related to $\delta\boldsymbol{q}$ according to Eq. (23):

$$\delta\boldsymbol{q} = \frac{1}{2}(\delta\boldsymbol{R}_\gamma + \delta\boldsymbol{R}_\gamma)\boldsymbol{R}_0[0 \quad 1 \quad 0]^{\mathrm{T}} = \frac{1}{2}(\delta\tilde{\boldsymbol{\theta}}_1^g\boldsymbol{q}_1 + \delta\tilde{\boldsymbol{\theta}}_2^g\boldsymbol{q}_2), \tag{39}$$

The expression of the matrix $\boldsymbol{G}_\theta$ can be obtained through Eq. (39) and $\boldsymbol{G}_\theta = \frac{\partial\boldsymbol{\theta}_r^e}{\partial\boldsymbol{P}_g^g}$. The detailed derivation can be found in reference (Crisfield, 1990). Eq. (37) yields the relationship between the force vector in the global coordinates and the internal force vector in the local coordinates.

$$\boldsymbol{F}^g = \boldsymbol{B}_a^{\mathrm{T}}\boldsymbol{f}_a, \tag{40}$$

Similarly, by considering the variation of the force vector in the global coordinates in Eq. (37), it can be obtained as follows:

$$\begin{cases} \delta\boldsymbol{F}^g = \boldsymbol{B}_a^{\mathrm{T}}\delta\boldsymbol{f}_a + \delta\boldsymbol{r}^{\mathrm{T}}\boldsymbol{f}_{a1} + \delta(\boldsymbol{E}\boldsymbol{P}^{\mathrm{T}})\boldsymbol{m} \\ \boldsymbol{m} = [f_{a2} \quad f_{a3} \quad f_{a4} \quad f_{a5} \quad f_{a6} \quad f_{a7}]^{\mathrm{T}} \end{cases}, \tag{41}$$

where $f_{ai}(i = 1, \cdots, 7)$ represent the components of the force vector $\boldsymbol{f}_a$. In conclusion, the tangent stiffness matrix in the global coordinate system can be obtained as follows:

$$\begin{cases} \boldsymbol{K}^g = \boldsymbol{B}_a^{\mathrm{T}}\boldsymbol{K}_a\boldsymbol{B}_a + \boldsymbol{K}_m \\ \boldsymbol{K}_m = \boldsymbol{D}f_{a1} - \boldsymbol{E}\boldsymbol{Q}\boldsymbol{G}_\theta{}^{\mathrm{T}}\boldsymbol{E}^{\mathrm{T}} + \boldsymbol{E}\boldsymbol{G}_\theta\boldsymbol{a}\boldsymbol{r}' \end{cases}, \tag{42}$$

where:

$$\boldsymbol{D} = \begin{bmatrix} \boldsymbol{d} & 0 & -\boldsymbol{d} & 0 \\ 0 & 0 & 0 & 0 \\ -\boldsymbol{d} & 0 & \boldsymbol{d} & 0 \\ 0 & 0 & 0 & 0 \end{bmatrix}, \boldsymbol{d} = \frac{1}{l}(\boldsymbol{I} - \boldsymbol{r}_1\boldsymbol{r}_1^{\mathrm{T}}), \tag{43}$$

$$\boldsymbol{Q} = \begin{bmatrix} \tilde{\boldsymbol{Q}}_1 \\ \tilde{\boldsymbol{Q}}_2 \\ \tilde{\boldsymbol{Q}}_3 \\ \tilde{\boldsymbol{Q}}_4 \end{bmatrix}, \boldsymbol{a} = \begin{bmatrix} 0 \\ \eta(f_{a2} + f_{a5})/l - (f_{a3} + f_{a6})/l \\ (f_{a4} + f_{a7})/l \end{bmatrix}, \tag{44}$$

$$\boldsymbol{P}^{\mathrm{T}}\boldsymbol{m} = [\boldsymbol{Q}_1^{\mathrm{T}} \quad \boldsymbol{Q}_2^{\mathrm{T}} \quad \boldsymbol{Q}_3^{\mathrm{T}} \quad \boldsymbol{Q}_4^{\mathrm{T}}]^{\mathrm{T}}, \tag{45}$$

[revised manuscript text omitted]

Fig. 8 shows the load–displacement response of the conical cantilever beam subjected to a vertically downward point load at

its free end. The results from the proposed co-rotational beam model are compared with the numerical solution obtained using the Runge–Kutta method from reference (Marjamäki and Mäkinen, 2009) and with finite element results reported by Nguyen (Nguyen, 2013). As evident in Fig. 8, the response predicted by the proposed method aligns closely with the Runge–Kutta solution and shows improved agreement compared to Nguyen's finite element results. This comparison validates the accuracy and effectiveness of the proposed co-rotational model in capturing large deformation behavior in conical cantilever beams with variable cross-sections.

[Figure]

**Figure 7.** Conical beam

[Figure]

**Figure 8.** Normalized moment and deformation in tapered beam

**4.3 Variable taper frame model**

A well-know benchmark frame structure (Manuel et al., 1968), shown in Fig. 9, is commonly used to assess the performance of nonlinear analysis methods. In its original configuration, members AB and BC possess constant stiffness. Building upon this example, Francisco(de Araujo et al., 2017) proposed a modified version by introducing variable stiffness to column AB, as illustrated in Fig. 10. The cross-sectional properties at points A and B for this modified configuration are provided as follows:

$$\begin{cases} I_{xA} = 14.76042 \times 10^{-8} m^4, I_{zA} = 17.04167 \times 10^{-8} m^4 \\ I_{xB} = 0.09375 \times 10^{-8} m^4, I_{zB} = 0.27083 \times 10^{-8} m^4 \\ \quad\quad S_A = 8.5 \times 10^{-4} m^2, S_B = 1.0 \times 10^{-4} m^2 \end{cases}, \tag{46}$$

Beam BC retains a constant rectangular cross-section with $S = 0.006$ m$^2$ and $I = 2 \times 10^{-8}$ m$^4$. The material properties for the entire frame are assumed to be homogeneous, with an elastic modulus $E = 7.2 \times 10^9$ GPa and Poisson's ratio $v=0.3$. The frame is discretized into 20 elements, and the interpolation method proposed in this study is applied. The resulting vertical and horizontal displacements at the load application points are compared with those obtained from Francisco2017 and a highly refined finite element mesh reported in reference (de Araujo et al., 2017). As shown in Fig. 11, the responses match quite well.

[Figure]

**Figure 9.** Frame scheme

[Figure]

Cross-section A    Cross-section B

**Figure 10.** Column geometry

[Figure]

(a) $u_x$ Displacement at node 13                    (b) $u_y$ Displacement at node 13

**Figure 11.** Frame displacement at node 13

**4.4 Natural frequencies of the conical cantilever beam**

This section considers an experimental conical cantilever beam reported in reference (Le et al., 2011) to verify the developed beam element. A modal analysis is performed where the natural frequencies are compared. The beam has a total length of 0.5m, with a fixed-end section diameter of 0.03m, and a free-end section diameter of $0.005m$. The mass density and elastic modulus are 7800kg/$m^3$ and 210 GPa, respectively. The specific experimental setup is described in detail in reference (Le et al., 2011). The first five natural frequencies of the conical beam are computed using the proposed variable cross-section beam model and are compared with both experimental results and two numerical approaches from Ref. (Le et al., 2011). The comparison is presented in Table 4.

**Table 4.** Natural frequencies of the conical cantilever beam

| Natural modes | TMM using Bessel functions (errors) [Hz] % | TMM using cylindrical elements (errors) [Hz] % | Present result (errors) [Hz] % | Experimental results [Hz] |
|---|---|---|---|---|
| Mode1 | 160.7(1.1) | 162.5(2.2) | 166.4(4.6) | 159.0 |
| Mode2 | 455.5(3.0) | 457.4(3.4) | 445.0(0.6) | 442.2 |
| Mode3 | 962.8(7.3) | 963.0(7.3) | 920.2(2.5) | 897.5 |
| Mode4 | 1702.0(6.9) | 1699.0(6.7) | 1658.9(4.2) | 1592.1 |
| Mode5 | 2679.1(7.0) | 2671.5(6.7) | 2607.8(4.1) | 2504.0 |

From Table 4, all three numerical methods produce results reasonably close to the experimental values. However, the proposed model demonstrates superior accuracy, with relative errors consistently below 5% across all five modes. In contrast, the relative errors of the TMM approaches in (Le et al.,2011) exceed 5% in several modes. Notably, the present model yields the most accurate results for the second and third modes, with relative errors of only 0.6% and 2.5%, respectively. These results confirm that the proposed variable cross-section beam model is effective in predicting the dynamic behavior of conical cantilever beams.

**4.5 3D frame structure with variation of beam cross-sections**

Figure 12 shows a 3D frame, with beams of varying circular cross-sections, loaded by concentrated loads F at node 1. The displacements of nodes 1 to 4 were founded. Variation of the cross-sectional area of the beams $a$ is defined by the following diameter quadratic function $d(y) = 0.04 + 0.04y^2$. The beams $b$ and $c$ have constant diameters through lengths of elements. Detailed parameters can be found in (Murín et al. 2002). Only one exact beam element was used to model each beam (a, b, c). In the Hermite beam element model, only one element was used to represent the beams b and c in all cases, but beams a were modelled with 1, 2 and 3 elements in models 1, 2 and 3 respectively.

[Figure]

**Figure 12.** Frame displacement at node 13 (**Murín et al. 2002**)

The numerical results obtained by the present method are compared against those from the method proposed by Murín et al. (2002) and the solutions from classical Hermite beam elements, as presented in Table 5. It can be observed from the table that compared to the reference method, the displacement solutions of the present method at all nodes and under all loading cases are consistently closer to the exact solution, demonstrating a significant enhancement in computational accuracy. Furthermore, when the number of elements is varied, the present method exhibits a narrower and more stable variation range in its solutions, highlighting its superior numerical robustness.

**Table 5.** Comparison of results

|  | Node1 (errors %) | | Node2 (errors %) | | Node3 (errors %) | | Node4 (errors %) | |
|---|---|---|---|---|---|---|---|---|
|  | $U_x$(mm) | $U_z$(mm) | $U_x$(mm) | $U_z$(mm) | $U_x$(mm) | $U_z$(mm) | $U_x$(mm) | $U_z$(mm) |
| Exact solution | 0.775 | -1.098 | 0.774 | -0.428 | 0.945 | -0.428 | 0.945 | -1.098 |
| Model1 ref | 0.651(16.0) | -0.882(19.7) | 0.650 | -0.336(21.5) | 0.763(19.3) | -0.336 | 0.763 | -0.882 |
| Model1 this paper | 0.745(3.9) | -0.981(10.7) | 0.745 | -0.427(0.2) | 0.859(9.1) | -0.427 | 0.859 | -0.981 |
| Model2 ref | 0.743(4.1) | -1.008(8.2) | 0.722 | -0.390(8.9) | 0.869(8.0) | -0.390 | 0.869 | -1.008 |
| Model2 this paper | 0.767(1.0) | -1.085(1.2) | 0.766 | -0.423(1.2) | 0.933(1.3) | -0.423 | 0.933 | -1.086 |
| Model3 ref | 0.749(3.4) | -1.054(4.0) | 0.748 | -0.409(4.4) | 0.908(3.9) | -0.409 | 0.908 | -1.054 |
| Model3 this paper | 0.772(0.4) | -1.093(0.5) | 0.771 | -0.426(0.5) | 0.940(0.5) | -0.426 | 0.940 | -1.093 |

**5.Conclusions**

This study proposes a novel co-rotational finite element framework for the geometrically nonlinear analysis of variable cross-section Timoshenko beams, which significantly enhances computational accuracy, efficiency, and robustness through the introduction of analytical displacement shape functions and a Gaussian integration strategy.

Case 1: A variable cross-section beam element based on analytical displacement shape functions is proposed, replacing traditional interpolation functions and significantly enhancing computational accuracy in geometrically nonlinear analysis. Example 1 (large deformation of a uniform cross-section cantilever beam) and Example 3 (beam with constant taper) validate the accuracy of this element in capturing bending deformation, especially under variable cross-section conditions where it demonstrates higher convergence accuracy compared to conventional piecewise uniform cross-section approaches.

Case 2: Gaussian integration is introduced within the corotational framework to compute element matrices, eliminating the need for repeated moment-of-inertia calculations at each cross-section and thereby improving computational efficiency. Example 4 (large deformation of a linearly tapered beam) shows that the method maintains accuracy while outperforms existing

variable cross-section corotational methods in computational efficiency; for cases with pronounced nonlinear taper, the method can be extended flexibly by incorporating additional sectional information.

Case 3: A local-to-global coordinate transformation method tailored for variable cross-section beams is developed, capable of handling irregular and proportionally graded sections, thus extending the applicability of the corotational formulation. Example 2 (spatial beam large-deformation experiment) and Example 5 (frame structure with varying taper) demonstrate that the method retains good numerical stability and robustness in three-dimensional large-deformation analysis and complex geometric nonlinear problems. Example 6 further confirms the framework's suitability for spatially tapered beam analysis.

In summary, the proposed corotational model for variable cross-section beams exhibits clear advantages in accuracy, efficiency, and generality, providing a reliable and efficient computational tool for predicting geometrically nonlinear responses of variable-section structures in engineering practice. Future work may focus on higher-order variable cross-section models, treatment of abruptly changing sections, and multiphysics coupling problems.

[revised manuscript text omitted]

Murín, Justín, and Vladimír Kutiš.: 3D-beam element with continuous variation of the cross-sectional area. Comput Struct 2002; 80(3-4): 329-338.